# Syntaxin 17 recruitment to mature autophagosomes is temporally regulated by PI4P accumulation

Saori Shinoda[1†], Yuji Sakai[1,2], Takahide Matsui[3], Masaaki Uematsu[1‡], Ikuko Koyama-Honda[1], Jun-ichi Sakamaki[1§], Hayashi Yamamoto[1,3], Noboru Mizushima[1*]

[1]Department of Biochemistry and Molecular Biology, Graduated School of Medicine, The University of Tokyo, Tokyo, Japan; [2]Department of Biosystems Science, Institute for Life and Medical Sciences, Kyoto University, Kyoto, Japan; [3]Department of Molecular Oncology, Institute for Advanced Medical Sciences, Nippon Medical School, Tokyo, Japan

*For correspondence: nmizu@m.u-tokyo.ac.jp

Present address: [†]Faculty of Life Sciences, Kyoto Sangyo University, Kyoto, Japan; [‡]Weill Institute for Cell and Molecular Biology, Cornell University, New York, United States; [§]Department of Physiology, Juntendo University Graduate School of Medicine, Tokyo, Japan

**Abstract** During macroautophagy, cytoplasmic constituents are engulfed by autophagosomes. Lysosomes fuse with closed autophagosomes but not with unclosed intermediate structures. This is achieved in part by the late recruitment of the autophagosomal SNARE syntaxin 17 (STX17) to mature autophagosomes. However, how STX17 recognizes autophagosome maturation is not known. Here, we show that this temporally regulated recruitment of STX17 depends on the positively charged C-terminal region of STX17. Consistent with this finding, mature autophagosomes are more negatively charged compared with unclosed intermediate structures. This electrostatic maturation of autophagosomes is likely driven by the accumulation of phosphatidylinositol 4-phosphate (PI4P) in the autophagosomal membrane. Accordingly, dephosphorylation of autophagosomal PI4P prevents the association of STX17 to autophagosomes. Furthermore, molecular dynamics simulations support PI4P-dependent membrane insertion of the transmembrane helices of STX17. Based on these findings, we propose a model in which STX17 recruitment to mature autophagosomes is temporally regulated by a PI4P-driven change in the surface charge of autophagosomes.

## eLife assessment

This paper addresses a **fundamental** issue in the field of autophagy: how is a protein responsible for autophagosome-lysosome fusion recruited to mature autophagosomes but not immature ones? The work succeeds in its ambition to provide a new conceptual advance. The evidence supporting the conclusions is **convincing**, with fluorescence microscopy, biochemical assays, and molecular dynamics simulations. This work will be of broad interest to cell biologists and biochemists studying autophagy, and also those focusing on lipid/membrane biology.

## Introduction

Macroautophagy (hereafter, autophagy) is a highly conserved process of intracellular degradation (*Mizushima and Levine, 2020*; *Søreng et al., 2018*). Membrane cisternae (called isolation membranes or phagophores) elongate, bend, and engulf cytoplasmic components. Closure of the rim of the cup-shaped structures results in the formation of autophagosomes. Subsequently, the autophagosomes fuse with lysosomes to degrade their enclosed contents. Autophagosome–lysosome fusion is strictly regulated; lysosomes fuse with only fully closed autophagosomes. If lysosomes were to fuse with

intermediate unclosed structures and their inner membrane is degraded, then harmful lysosomal enzymes would leak out into the cytosol. This temporal regulation of autophagosome–lysosome fusion is achieved in mammals by two mechanisms. One is the regulated translocation of the autophagosomal SNARE syntaxin 17 (STX17), which is recruited immediately before or after the closure of autophagosomes (*Itakura et al., 2012*; *Takáts et al., 2013*; *Tsuboyama et al., 2016*). The other is the activation of YKT6, a conserved autophagosomal SNARE (*Matsui et al., 2018*), which is kept inactivated by ULK1 (a homolog of yeast Atg1)-mediated phosphorylation until the completion of autophagosome formation (ULK1/Atg1 is dissociated at this time; *Barz et al., 2020*; *Essmann et al., 1995*; *Sánchez-Martín et al., 2023*). Then, STX17 interacts with SNAP29 and the lysosomal SNARE protein VAMP7 or VAMP8, and YKT6 interacts with SNAP29 and the lysosomal STX7 to mediate the fusion between autophagosomes and lysosomes.

However, the molecular mechanism underlying the late recruitment of STX17 is not known. A previous study reported that LC3/GABARAP family proteins and immunity-related GTPase M (IRGM) are involved in autophagosomal recruitment of STX17 (*Kumar et al., 2018a*), whereas other studies have shown that LC3/GABARAP family proteins are not required (*Nguyen et al., 2016*; *Tsuboyama et al., 2016*). It has also been recently reported that phosphorylation of STX17 and binding to filamin A are important for STX17 recruitment (*Wang et al., 2023*). Although these factors may help STX17 recruitment, these mechanisms do not explain the strict temporal regulation of STX17 recruitment.

Accordingly, we hypothesized that there exists a yet unknown change in some autophagosomal properties during maturation that is recognized by STX17. In this study, we reveal that STX17 recruitment to autophagosomes requires positively charged amino acids in the C-terminal region of STX17. Consistently, the membrane of autophagosomes becomes more negatively charged when autophagosomes acquire STX17. Furthermore, we show that phosphatidylinositol 4-phosphate (PI4P), a negatively charged phospholipid, accumulates during autophagosome maturation and is required for STX17 recruitment. Based on these findings, we propose a model in which STX17 recruitment to mature autophagosomes is temporally regulated by the electrostatic change of autophagosomes.

## Results

### Autophagosomal localization of STX17 requires a positively charged C-terminal region

STX17 has an N-terminal Habc domain, followed by a SNARE domain, two tandem transmembrane helices, and a short C-terminal region, with both N- and C-terminal ends facing the cytosol (*Figure 1A*; *Itakura et al., 2012*). A short construct containing only the transmembrane helices and the C-terminal region (STX17TM), which behaves similarly to full-length STX17 (*Itakura et al., 2012*), colocalized with ring-shaped autophagosomes labeled with the general autophagic membrane marker microtubule-associated protein light chain 3B (LC3B; *Figure 1B and C*). STX17 is a tail-anchored protein, and the C-terminal region of tail-anchored proteins is generally important for specific membrane targeting (*Rao et al., 2016*; *Yagita et al., 2013*). Therefore, we determined the role of the C-terminal region of STX17. In the following experiments, we used the STX17TM construct in order to avoid detecting an indirect effect of SNARE domain-mediated translocation. Deletion of the C-terminal cytosolic region from STX17TM (TMΔC) resulted in a diffuse cytosolic pattern (*Figure 1C*), indicating that not only the transmembrane helices but also the C-terminal region of STX17 are required for its proper autophagosomal localization. Although the role of STX17 in autophagosome–lysosome fusion is conserved in both *Drosophila melanogaster* (Dm; *Takáts et al., 2013*) and *Caenorhabditis elegans* (Ce; *Guo et al., 2014*), the amino acid sequence of the C-terminal region is not conserved in these organisms (*Figure 1—figure supplement 1A*). Nevertheless, when expressed in mammalian cells, DmSTX17TM and CeSTX17TM were recruited to autophagosomes, although less efficiently (*Figure 1—figure supplement 1B*), suggesting that recruitment of STX17 does not depend on the specific amino acid sequence in the C-terminal region.

In general, cationic amino acids in the C-terminal region of tail-anchored proteins are important for targeting membranes (*Borgese et al., 2001*; *Horie et al., 2002*; *Yagita et al., 2013*). STX17 has cationic amino acids in its C-terminal region, and the replacement of lysine and arginine with alanine impaired autophagosomal localization in a dose-dependent manner (*Figure 1A and D*, *Figure 1—figure supplement 1C*). Moreover, the replacement of the C-terminal region with artificial sequences

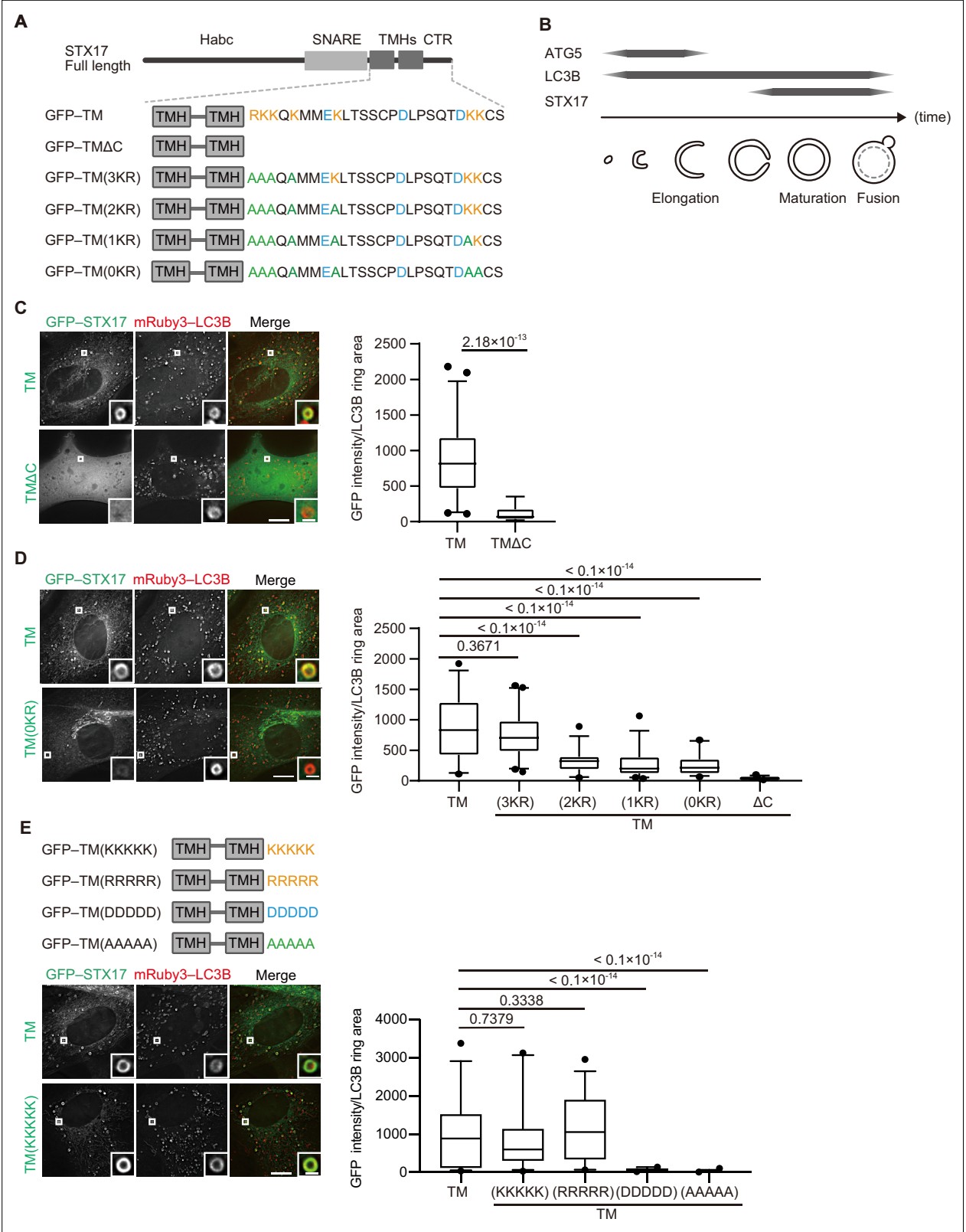

**Figure 1.** The recruitment of STX17 to autophagosomes is dependent on its positively charged C-terminal region. (**A**) Schematic representation of the structures of STX17 and its C-terminal variants. The positively (orange) and negatively (blue) charged residues are shown. Alanine substitutions are shown in green. TMH, transmembrane helix; CTR, C-terminal region. (**B**) Schematic representation of the localization of ATG5, LC3B, and STX17 during autophagosome formation and maturation. (**C–E**) Mouse embryonic fibroblasts (MEFs) stably expressing mRuby3-LC3B and GFP–STX17TM (containing

*Figure 1 continued on next page*

*Figure 1 continued*

the two transmembrane helices and the C-terminal region) or its mutants were cultured in starvation medium for 1 hr. Quantification of GFP–STX17TM intensity of mRuby3–LC3B-positive ring-like structures (n>30) are shown in the graphs. In box plots, solid horizontal lines indicate medians, boxes indicate the interquartile ranges (25th to 75th percentiles), whiskers indicate the 5th to 95th percentiles, and dots represent outliers. Differences were statistically analyzed by Welch's *t*-test (**C**) or one-way ANOVA followed by Dunnett's multiple comparison test (**D and E**). Experiments were performed three times independently. Scale bars, 10 µm (main), 1 µm (inset) (**C, D, and E**).

The online version of this article includes the following source data and figure supplement(s) for figure 1:

**Source data 1.** Data used for graphs presented in *Figure 1C, D and E* and *Figure 1—figure supplement 1B*.

**Figure supplement 1.** Recruitment of STX17 depends on the abundance of cationic amino acids in the C-terminal region but not on its specific amino acid sequence.

consisting of five residues of positively charged lysine or arginine, but not negatively charged aspartic acid or uncharged alanine, restored the localization of STX17 to autophagosomes (*Figure 1E*, *Figure 1—figure supplement 1D*). These data suggest that autophagosomal localization of STX17 requires positively charged residues, but not sequence-specific structures, in its C-terminal region.

## The membrane of autophagosomes becomes negatively charged during maturation

Given the importance of positively charged residues in the C-terminal region, we hypothesized that STX17 favors negatively charged membranes. To evaluate this hypothesis, we first conducted an in vitro membrane binding assay using liposomes with different compositions of phospholipids. STX17TM was efficiently recruited to liposomes containing anionic phospholipids such as phosphatidylserine (PS), phosphatidylinositol (PI) 3-phosphate (PI3P) or PI4P (*Figure 2A*). The association of STX17TM with PI4P-containing membranes was abolished in the presence of 1 M NaCl (*Figure 2B*). These data suggest that STX17 can be recruited to negatively charged membranes via electrostatic interaction independent of the specific lipid species.

Next, we tested whether autophagosomal membranes are indeed negatively charged. To monitor membrane charge in vivo, we used membrane surface charge biosensors (*Figure 2C*; *Simon et al., 2016*; *Yeung et al., 2006*). These surface charge probes have a diverse range of positively charged peptides and a farnesylated anchor at their C terminus, with GFP fused to their N terminus. The name of each probe is indicated by the number of lysine (positively charged) followed by that of glutamine. The probe for the most negatively charged membrane (9K0Q) labeled the plasma membrane, which is known to have a highly negative charge (*Li et al., 2014*), but not autophagosomes (*Figure 2D and E*, *Figure 2—figure supplement 1A*). However, the probes for intermediate (5K4Q and 3K6Q) and weakly (1K8Q) negative charges labeled not only the plasma membrane but also LC3B- and STX17-positive autophagosomes (*Figure 2D and E*, *Figure 2—figure supplement 1A*). The recruitment kinetics differed between the intermediate and weakly negative charge probes. The 1K8Q probe was recruited to not only STX17-positive spherical autophagosomes but also STX17-negative, LC3B-positive elliptic structures that should correspond to unclosed autophagosomes (*Tsuboyama et al., 2016*; *Figure 2F*). In contrast, the intermediate charge probes (3K6Q and 5K4Q) were recruited to spherical LC3B-positive autophagosomes at almost the same time as STX17 (*Figure 2G*, *Figure 2—figure supplement 1B*). Structures positive for ATG5, a marker for unclosed autophagosomes, were labeled with 1K8Q but not with 3K6Q or 5K4Q (*Figure 2—figure supplement 1C*). These results, which are summarized in *Figure 2H*, suggest that mature autophagosomes are more negatively charged compared with unclosed intermediate structures.

## The accumulation of PI4P in mature autophagosomes coincides with STX17 recruitment

We next explored the mechanism that generates the negative charges of autophagosomal membranes. As the change in the membrane charge was rapid and robust, we reasoned that it could be caused by a change in lipid composition or modification. We screened negatively charged lipids that are enriched in autophagosomal membranes using various phospholipid probes (*Platre and Jaillais, 2016*). Among them, we found that the probes for PI4P GFP-fused PH domain of CERT; GFP–CERT(PHD) and phosphatidylinositol 3,5-bisphosphate (PI(3,5)P$_2$) (GFP–TRPML1(PHD)) colocalized with STX17-positive

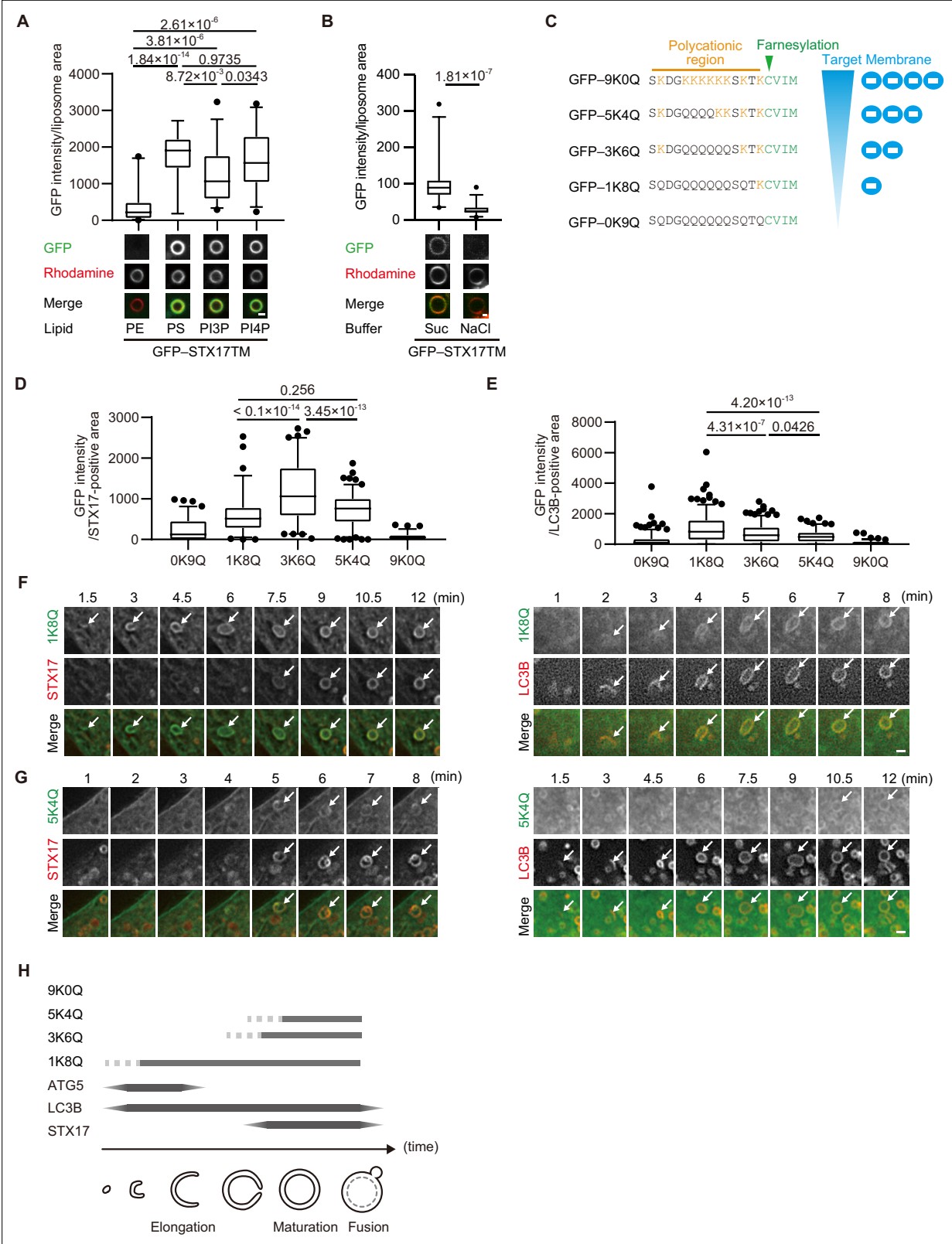

**Figure 2.** The membrane of autophagosomes becomes negatively charged during maturation. (**A**) GFP–STX17TM translated in vitro was incubated with rhodamine-labeled liposomes containing the indicated concentrations of phospholipids: 70% phosphatidylcholine (PC), 20% phosphatidylethanolamine (PE), and 10% of either PE, phosphatidylserine (PS), phosphatidylinositol 3-phosphate (PI3P), or phosphatidylinositol 4-phosphate (PI4P). GFP intensities of liposomes are quantified and shown as in **Figure 1C** (n>30). (**B**) GFP–STX17TM translated in vitro was incubated with rhodamine-labeled liposomes

*Figure 2 continued on next page*

*Figure 2 continued*

containing 70% PC, 20% PE and 10% PI4P in the presence of 1 M NaCl or 1.2 M sucrose. GFP intensities of liposomes were quantified and shown as in *Figure 1C* (n>30). (**C**) Amino acid sequences of GFP-tagged membrane surface charge probes. The positively charged residues are shown in orange. The farnesylation motif is indicated in green. (**D and E**) Mouse embryonic fibroblasts (MEFs) stably expressing one of the GFP-tagged charge probes and mRuby3–STX17TM (**C**) or mRuby3–LC3B (**D**) were cultured in starvation medium for 1 hr. GFP intensities of mRuby3–STX17TM-positive (**C**) or mRuby3–LC3B-positive (**D**) ring-like structures were quantified (n>70). (**F and G**) Time-lapse analysis of MEFs stably expressing the GFP-tagged 1K8Q (**E**) or 5K4Q (**F**) charge probes and mRuby3–STX17TM or mRuby3–LC3B cultured in starvation medium. Autophagosomes are indicated by arrows. (**H**) Summary of electrostatic dynamics of autophagosome formation. In box plots, solid horizontal lines indicate medians, boxes indicate the interquartile ranges (25th to 75th percentiles), whiskers indicate the 5th to 95th percentiles, and dots represent outliers. Differences were statistically analyzed by Welch's *t*-test (**B**) or one-way ANOVA followed by Sidak's multiple comparison test (**A, C, and D**). Experiments were performed three times independently. Scale bars, 1 µm (**A, B, F, and G**).

The online version of this article includes the following source data and figure supplement(s) for figure 2:

**Source data 1.** Data used for graphs presented in *Figure 2A, B, D and E*.

**Figure supplement 1.** The membrane of autophagosomes becomes negatively charged during maturation.

ring-shaped structures (*Figure 3—figure supplement 1A*). Since PI(3,5)P$_2$ was primarily present in lysosomes (*Figure 3—figure supplement 1B*; *Dong et al., 2010*), PI(3,5)P$_2$ enrichment likely occurs after fusion with lysosomes. The CERT(PHD) (W33A) mutant, lacking PI4P-binding activity, was not co-localized with STX17TM (*Figure 3—figure supplement 2A*; *Sugiki et al., 2012*). Consistently, other PI4P probes, including FAPP(PHD), OSBP(PHD), and P4M-SidMx2, colocalized with STX17TM (*Figure 3—figure supplement 2B, C*; *Hammond et al., 2014*; *Platre and Jaillais, 2016*).

PI4P is an important lipid component of the Golgi complex and the plasma membrane as well as a major regulator of conserved eukaryotic cellular processes (*Balla, 2013*; *Platre et al., 2018*; *Schink et al., 2016*). In the context of autophagy, it was previously reported that GABARAP-dependent PI4P production by PI4K2A (PI4KIIα) is required for autophagosome–lysosome fusion (*Wang et al., 2015*), and that PI4KB (PI4KIIIβ) is detected in ATG9A vesicles and early autophagic structures and is furthermore required for autophagosome formation (*Judith et al., 2019*). Consistently, we detected faint signals of PI4K2A and PI4KB on LC3B-positive structures (*Figure 3—figure supplement 1C*). However, the dynamics of PI4P production during autophagosome maturation has not been systematically determined.

The PI4P probes (CERT(PHD), FAPP(PHD), OSBP(PHD), and P4M-SidMx2) colocalized with STX17 but not with ATG5 (*Figure 3A*, *Figure 3—figure supplement 2B, C*), suggesting that PI4P accumulates in mature autophagosomes. We then investigated the dynamics of PI4P enrichment by time-lapse microscopy. The signal of the CERT(PHD) probe appeared on LC3B-positive membranes only after the disappearance of the unclosed autophagosome markers ATG5 and WIPI2B (*Figure 3B and C*) and almost at the same time as STX17 localized (*Figure 3D*). This was followed by the appearance of LysoTracker signals (note that LysoTracker signals have been previously observed to appear ring-shaped until inner membrane degradation *Tsuboyama et al., 2016*; *Figure 3—figure supplement 2D*). To confirm that PI4P accumulation was independent of autophagosome–lysosome fusion, we evaluated the colocalization among the PI4P probe, LC3B, and LAMP1 in STX17- and YKT6-double-knockdown cells, in which unfused autophagosomes accumulated (*Matsui et al., 2018*). The punctate structures of CERT(PHD) were still colocalized with LC3B-positive and LAMP1-negative structures in these double-knockdown cells (*Figure 3—figure supplement 2E*). In addition, STX17 recruitment and PI4P enrichment occurred normally in cells lacking all ATG8 family proteins (LC3A, LC3B, LC3C, GABARAP, GABARAPL1, and GABARAPL2) (*Figure 3—figure supplement 2F*). These results suggest that PI4P accumulates in mature autophagosomal membranes independent of lysosome fusion and ATG8 proteins, and that the kinetics of the accumulation of STX17, the PI4P probes, and the negative charge probes are correlated.

## STX17 recruitment to autophagosomes depends on PI4P

To determine whether PI4P is required for the recruitment of STX17 to autophagosomes, we first tried to dephosphorylate PI4P by ectopic expression of SAC1, a PI4P-phosphatase, on autophagosomes (*Manford et al., 2010*). To this end, we fused the phosphatase domain of yeast Sac1 (Sac1PD) to the N terminus of LC3B and expressed them in different cell lines, using several methods, including lipofection and retrovirus- and adenovirus-mediated transfection. Although Sac1PD–LC3B localized

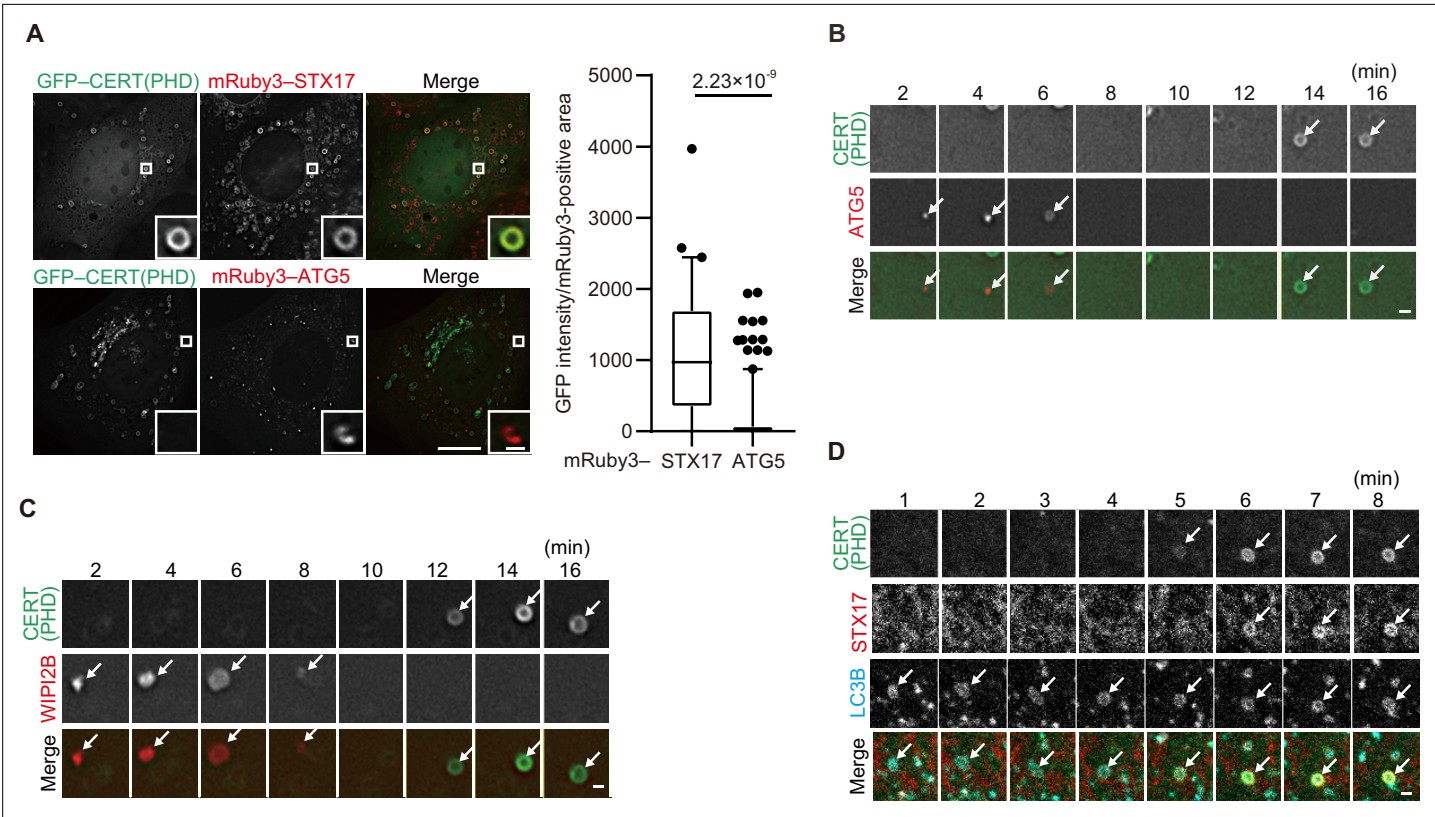

**Figure 3.** Phosphatidylinositol 4-phosphate (PI4P) is enriched in the autophagosomal membrane during maturation. (**A**) Mouse embryonic fibroblasts (MEFs) stably expressing GFP–CERT(PHD) and mRuby3–STX17TM or mRuby3–ATG5 were cultured in starvation medium for 1 hr. GFP intensities of mRuby3-positive structures (n>60) were quantified. In box plots, solid horizontal lines indicate medians, boxes indicate the interquartile ranges (25th to 75th percentiles), whiskers indicate the 5th to 95th percentiles, and dots represent outliers. Differences were statistically analyzed by Welch's *t*-test. (**B–D**) Time-lapse analysis of MEFs stably expressing GFP–CERT(PHD) and mRuby3–ATG5 (**B**), WIPI2B–mRuby3 (**C**), or mRuby3–STX17TM and HaloTag–LC3B (visualized with SaraFluor 650T HaloTag ligand) (**D**) cultured in starvation medium. Autophagosomes are indicated by arrows. Experiments were performed three times independently. Scale bars, 10 µm (**A** [main]), 1 µm (**A** [inset], **B–D**).

The online version of this article includes the following source data and figure supplement(s) for figure 3:

**Source data 1.** Data used for graphs presented in *Figure 3A* and *Figure 3—figure supplement 2C*.

**Figure supplement 1.** Localization of phospholipids in mature autophagosomes.

**Figure supplement 2.** Phosphatidylinositol 4-phosphate (PI4P) is enriched in mature autophagosomes before fusion with lysosomes.

to autophagosomes, it did not reduce either the level of autophagosomal PI4P or STX17 recruitment (unpublished observation). High expression of Sac1PD–LC3B inhibited autophagosome formation (unpublished observation). Mammals possess four PI 4-kinases. We next tried to deplete each or combinations of these PI 4-kinases by using siRNA or an auxin-inducible degron system (*Yesbolatova et al., 2020*) or inhibit them using PI 4-kinase inhibitors, including PI-273 (*Li et al., 2017*), BF738735 (unpublished observation; *van der Schaar et al., 2013*), and NC03 (*Figure 4—figure supplement 1*; *Sengupta et al., 2019*), but ultimately failed to reduce the autophagosomal PI4P levels and STX17 recruitment. Although the precise reason for this failure was not known, these approaches were insufficient to deplete autophagosomal PI4P in vivo.

Therefore, we instead used an in vitro system to determine whether PI4P is important for STX17 recruitment to autophagosomes (*Figure 4A*). Mature autophagosomes prior to fusion with lysosomes were isolated from *STX17* knockout (KO) cells (*Figure 4B*, Fraction #1; *Matsui et al., 2018*). Recombinant Sac1PD, its phosphatase-dead mutant (C392S), and mGFP–STX17TM were generated using insect cells (*Figure 4C*). Autophagosomes were first incubated with or without Sac1PD or Sac1PD (C392S) for 30 min and then further incubated with mGFP–STX17TM for another 30 min. Recombinant mGFP–STX17TM was associated with autophagosomes, but it was significantly impaired by treatment

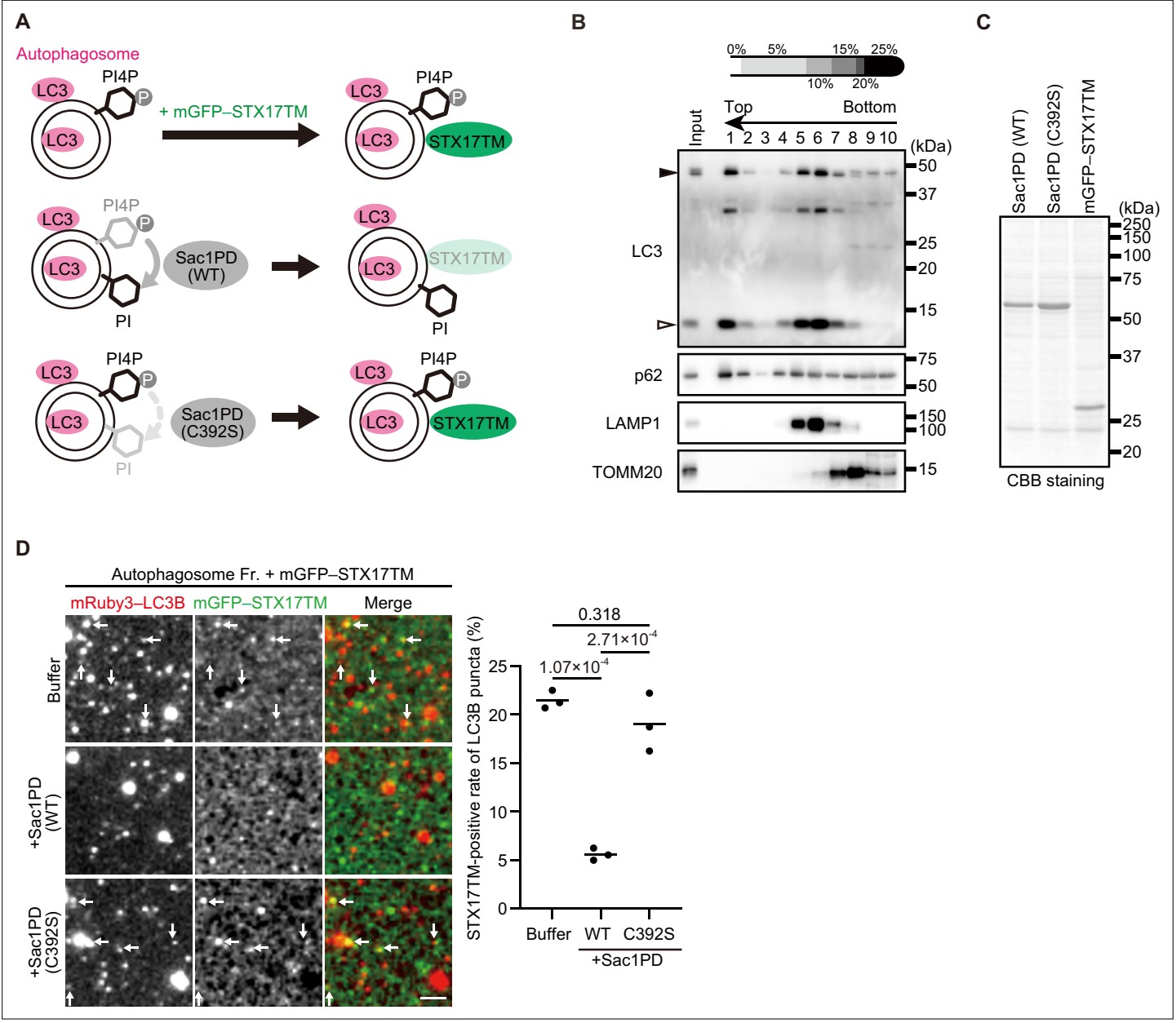

**Figure 4.** STX17 recruitment to autophagosomes depends on phosphatidylinositol 4-phosphate (PI4P) in vitro. (**A**) Schematic representation of the in vitro autophagosome recruitment assay. Isolated autophagosomes were mixed with mGFP–STX17TM and either recombinant Sac1-phosphatase domain (Sac1PD) or its phosphatase-dead mutant (C392S). (**B**) Isolation of mature autophagosomes prior to their fusion with lysosomes. Homogenates of *STX17* knockout HeLa cells stably expressing mRuby3–LC3B cultured in starvation medium at 1 hr were separated by the OptiPrep membrane flotation method. The autophagosome-containing fraction (#1: LC3-positive and LAMP1-negative) was collected. The positions of mRuby3–LC3B (black arrowhead) and endogenous LC3B (white arrowhead) are indicated. (**C**) Purification of recombinant yeast Sac1 (phosphatase domain, PD) and its phosphatase-dead (C392S) mutant and mGFP–STX17TM from High Five cells. (**D**) In vitro autophagosome association assay. Isolated autophagosomes were mixed with recombinant Sac1 (WT or C392S) for 30 min and then with mGFP–STX17TM for another 30 min. Representative images are shown. STX17-positivity rates were determined across three independent experiments (two of the three experiments were performed in a blind manner, and 80 autophagosomes were counted in each experiment). Solid horizontal lines indicate means. Differences were statistically analyzed by one-way ANOVA followed by Tukey's test. The scale bar, 2.5 μm.

The online version of this article includes the following source data and figure supplement(s) for figure 4:

**Source data 1.** Data used for graphs presented in *Figure 4D*, *Figure 4—figure supplement 1A, B*.

**Source data 2.** Uncropped blot images of *Figure 4B and C*.

**Figure supplement 1.** The PI 4-kinase inhibitor NC03 failed to suppress autophagosomal PI4P accumulation and STX17 recruitment.

with Sac1PD. Phosphatase-dead Sac1PD (C392S) showed no effect (*Figure 4D*). These data suggest that PI4P is important for autophagosomal recruitment of STX17.

We further determined the effect of PI4P on the dynamics of STX17TM on a membrane by performing a molecular dynamics simulation. We used all-atom models for STX17TM and the highly mobile membrane-mimetic (HMMM) model for the lipid bilayer (*Ohkubo et al., 2012*). The structure of STX17TM was predicted by trRosetta (*Du et al., 2021*), yielding five different models, all of which were used in the simulation. The initial configuration was prepared such that the center of mass of STX17TM was located 3 nm above the membrane surface, after which the movement of STX17TM was tracked. In independent simulations using four out of the five predicted STX17TM structures, STX17TM was inserted into the membrane with PI4P (PC:PE:PI4P=70:20:10) within a short time scale of 100 ns (*Figure 5A and B*, *Figure 5—video 1*). The two transmembrane helices were inserted into the membrane, while the charged C-terminal region remained bound to the membrane surface (*Figure 5—video 1*). In contrast, STX17 diffused freely in the solution and was not inserted into the membrane without PI4P (PC:PE = 70:30) in simulations of all five structures (*Figure 5C and D*, *Figure 5—video 2*). Moreover, if the membrane contained PI instead of PI4P, STX17TM approached the PI-containing membrane but was not inserted into the membrane (*Figure 5E and F*, *Figure 5—video 3*). These results suggest that STX17TM can be readily inserted into PI4P-containing membranes.

## Discussion

Based on the results of this study, we propose a model in which temporally regulated autophagosome–lysosome fusion involves a dynamic change in electrostatic status during autophagosome maturation; the recruitment of STX17 to mature autophagosomes is primarily mediated by the electrostatic interaction between the positively charged C-terminal region of STX17 and negatively charged autophagosomal membranes likely due to accumulation of PI4P (*Figure 5G*). During revising this manuscript, Juhász's group also reached a consistent model that PI4P is required for autophagosomal localization of STX17 (*Laczkó-Dobos et al., 2024*). Because STX17 does not localize to all negatively charged membranes (e.g. the plasma membrane), the transmembrane helices appear to also regulate the specificity of target membranes. This is consistent with the 'coincidence detector' mechanism in which both a cationic motif and a nearby membrane-anchored moiety are important for membrane targeting (*Carlton and Cullen, 2005*; *Yeung and Grinstein, 2007*).

A striking finding of this study is that the membrane of autophagosomes become more negatively charged during maturation, most likely through an increase in the PI4P content of their membranes, although we do not exclude the possible involvement of other negatively charged molecules. Given the possibility that fluorescence lipid probes may give false-negative results, a more comprehensive biochemical analysis, such as lipidomics analysis of mature autophagosomes, would be imperative to elucidate the potential involvement of other negatively charged lipids. Although PI3P is important for autophagosome formation, we do not think that PI3P contributes to the increase in negative charge. It was reported that PI3P is dephosphorylated to PI during autophagosome formation or maturation (*Allen et al., 2020*; *Cebollero et al., 2012*; *Taguchi-Atarashi et al., 2010*), and that the PI3P effectors DFCP1 and WIPI family proteins are not detected on mature autophagosomes (*Koyama-Honda et al., 2013*). In fact, we did not detect WIPI2B and the PI3P reporter GFP–2×FYVE on PI4P-enriched structures (*Figure 3C*) and STX17-positive structures (*Figure 3—figure supplement 1A*), respectively. Considering the change in autophagosomal membrane charge, the amount of PI4P produced during autophagosome maturation is likely to be higher than that of PI3P in the early stages. Our data of the 1K8Q probe suggest that immature autophagosomal membranes may also have slight negative charges (*Figure 2F*). Although the source of the negative charge of immature autophagosomes is currently unknown, it may be derived from low levels of PI4P, which is undetectable by the PI4P probes and/or other negatively charged lipids such as PI and PS (*Schmitt et al., 2022*).

PI4P is involved in many cellular processes, including autophagosome formation and maturation. It was reported that PI4KB regulates the early stage of autophagosome formation through interaction with ATG9A (*Judith et al., 2019*). It was also reported that PI4K2A is recruited to autophagosomes by binding to the ATG8 family proteins GABARAP and GABARAPL1 and produces PI4P in the autophagosomal membranes (*Wang et al., 2015*). Furthermore, the knockdown of PI4K2A resulted in the accumulation of abnormally enlarged autophagosomes, presumably owing to a defect in autophagosome–lysosome fusion (*Wang et al., 2015*). Indeed, a freeze-fracture replica labeling study

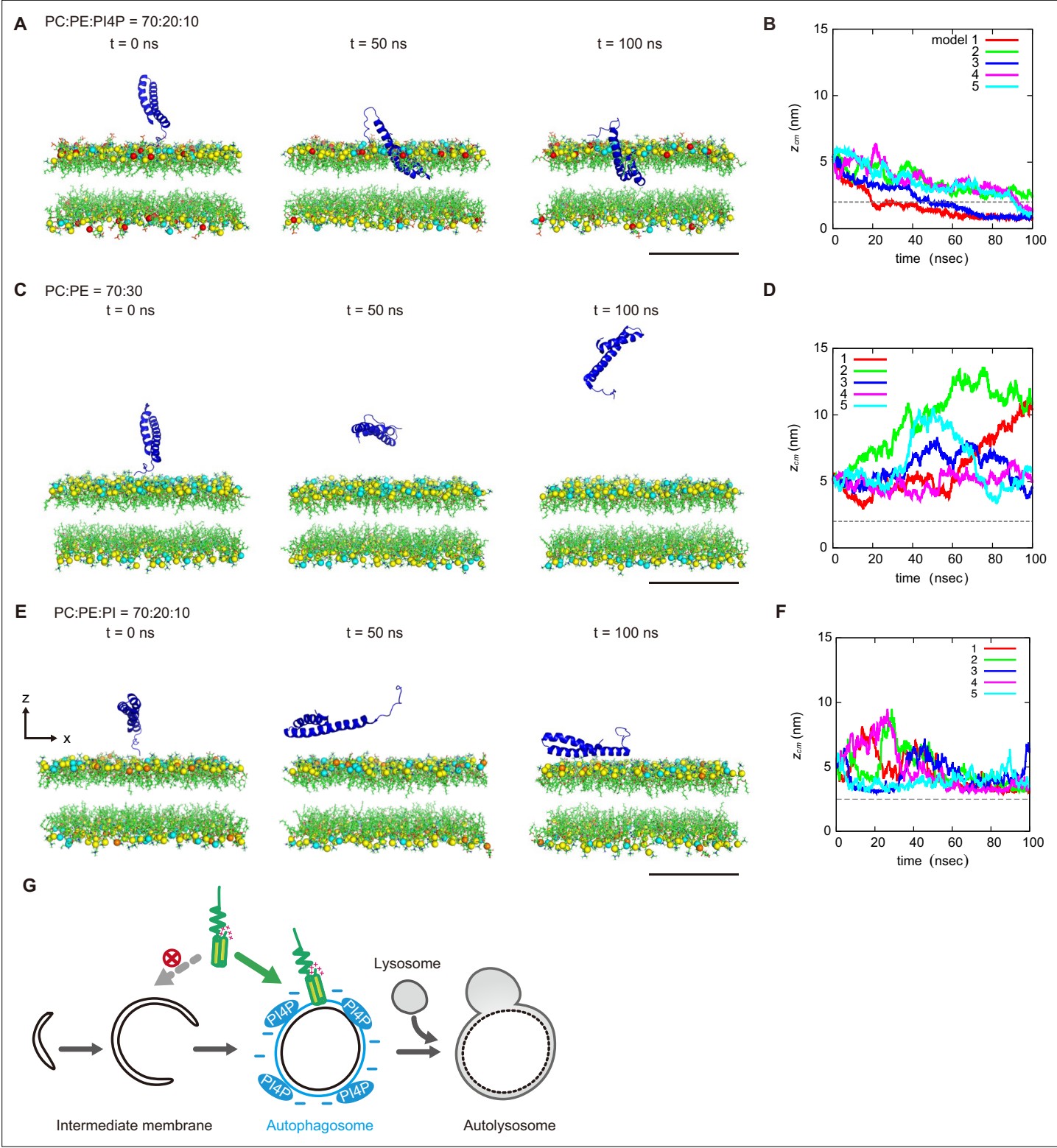

**Figure 5.** Molecular dynamics simulations of phosphatidylinositol 4-phosphate (PI4P)-dependent STX17TM insertion into membranes. (**A, C and E**) An example of a time series of simulated results of STX17TM insertion into a membrane consisting of 70% phosphatidylcholine (PC), 20% phosphatidylethanolamine (PE), and 10% PI4P (POPI14) (**A**), 70% PC and 30% PE (**C**) or 70% PC, 20% PE and 10% phosphatidylinositol (PI) (**E**). STX17TM is shown in blue. Phosphorus in PC, PE, PI4P, and PI are indicated by yellow, cyan, red, and orange, respectively. Short-tailed lipids are represented as green sticks. The time evolution series of (**A**), (**C and E**) are shown in *Figure 5—videos 1–3*. (**B, D and F**) Time evolution of the z-coordinate of the center of mass ($z_{cm}$) of the transmembrane helices of STX17TM in the case of membranes with PI4P (**B**) or PI (**F**) and without PI4P or PI (**D**). Five

*Figure 5 continued on next page*

*Figure 5 continued*

independent simulation results are represented by solid lines of different colors. The gray dashed lines indicate the locations of the lipid heads. Scale bars, 5 nm (**A, C, and E**). (**G**) Model of the PI4P-driven electrostatic maturation of the autophagosome and STX17 recruitment.

The online version of this article includes the following video and source data for figure 5:

**Source data 1.** Data used for graphs presented in *Figure 5B, D and F*.

**Figure 5—video 1.** Molecular dynamics simulations of STX17TM insertion into a phosphatidylinositol 4-phosphate (PI4P)-containing membrane.
https://elifesciences.org/articles/92189/figures#fig5video1

**Figure 5—video 2.** Molecular dynamics simulations of STX17TM insertion into a membrane without phosphatidylinositol 4-phosphate (PI4P).
https://elifesciences.org/articles/92189/figures#fig5video2

**Figure 5—video 3.** Molecular dynamics simulations of STX17TM insertion into a phosphatidylinositol (PI)-containing membrane.
https://elifesciences.org/articles/92189/figures#fig5video3

showed that autophagosomal membranes contain PI4P in the cytoplasmic leaflets of the outer and inner membranes (*Kurokawa et al., 2019*). Very recently, it was reported that the knockdown of PI4KIIα in *Drosophila* blocked autophagic flux at a late step of autophagy in the fat body (*Liu et al., 2023*). Although these studies are consistent with our finding that PI4P production induces STX17 recruitment for fusion with lysosomes, our repeated attempts to deplete PI4P genetically (i.e. by depletion of PI 4-kinases or attachment of Sac1 to autophagosomes) or pharmacologically did not deplete autophagosomal PI4P or inhibit STX17 recruitment. Liu et al. also showed that the knockdown of PI4KIIα in *Drosophila* did not inhibit STX17 recruitment. This may be because PI 4-kinases function redundantly. In any case, it would be important to elucidate how PI 4-kinase activity or PI4P synthesis is upregulated during autophagosome maturation.

Surface charge, which is now recognized as one of the key physical properties of cellular membranes, regulates various biological processes (*Yeung and Grinstein, 2007*). For example, the negatively charged plasma membrane, where anionic phospholipids such as $PI(4,5)P_2$ and PS are enriched, recruits positively charged proteins, including K-Ras and Rac1 (*Bigay and Antonny, 2012*; *Heo et al., 2006*; *Li et al., 2014*; *Platre et al., 2018*; *Yeung et al., 2008*; *Yeung et al., 2006*), some of which are released from the membrane when the negative charge of the surface is reduced during phagocytosis (*Yeung et al., 2006*). However, little has been reported on the role of electrostatic properties of other organelles, including autophagosomes.

Generally, PI4P can be structurally recognized by a series of PI4P-binding domain-containing proteins (*Laczkó-Dobos et al., 2024*; *Moravcevic et al., 2012*). However, recent studies suggest that PI4P can also generate a negatively charged electrostatic field and recruit positively charged proteins independent of specific amino acid motifs (*Hammond et al., 2012*; *Simon et al., 2016*). Examples of the role of PI4P in electrostatic signature include the recruitment of the polar auxin transport regulator PINOID and other signaling molecules to the cell plate in plants (*Simon et al., 2016*) the recruitment of D6 protein kinase (*Barbosa et al., 2016*), NPH3 (*Reuter et al., 2021*), and the exocyst complex (*Synek et al., 2021*) to the plasma membrane in plants; the recruitment of Lgl and Dlg proteins, which function in cell polarity and tumorigenesis, to the plasma membrane in *Drosophila* (*Darden et al., 1993*; *Lu et al., 2021*); and the recruitment of NLRP3 to the disassembled trans-Golgi network during inflammasome activation in mammals (*Chen and Chen, 2018*). In the present study, we showed that autophagosomal PI4P also serves as a regulator of surface charge. Given that the function of elongating unclosed autophagosomes (i.e. sequestration) and mature closed autophagosomes (i.e. fusion with lysosomes) differs despite using seemingly the same membrane, electrostatic maturation would be an efficient way for the function to switch.

# Materials and methods

## Key resources table

| Reagent type (species) or resource | Designation | Source or reference | Identifiers | Additional information |
|---|---|---|---|---|
| Gene (*H. sapiens*) | STX17 | https://doi.org/10.1016/j.cell.2012.11.001 | | |

*Continued on next page*

*Continued*

| Reagent type (species) or resource | Designation | Source or reference | Identifiers | Additional information |
|---|---|---|---|---|
| Gene (*D. melanpgaster*) | STX17 | NCBI Reference Sequence | NM_079202 | |
| Gene (*C. elegans*) | STX17 | NCBI Reference Sequence | NM_059941 | |
| Gene (*R. norvegicus*) | LC3B | https://doi.org/10.1083/jcb.200712064 | | |
| Gene (*M. musculus*) | ATG5 | NCBI Reference Sequence | NM_053069 | |
| Gene (*H. sapiens*) | 2×Spo20(PABD) | https://doi.org/10.1074/jbc.M116.742346 | NM_005633; amino acids 422–551 | For GFP tagged phospholipid probes |
| Gene (*H. sapiens*) | CERT(PHD) | https://doi.org/10.1007/s11010-005-9044-z | NM_001130105; amino acids 1–116 | For GFP tagged phospholipid probes |
| Gene (*H. sapiens*) | FAPP | https://doi.org/10.1091/mbc.e04-07-0578 | NM_001807; amino acids 1–101 | For GFP tagged phospholipid probes |
| Gene (*H. sapiens*) | OSBP | https://doi.org/10.1016/s0960-9822(98)70296-9 | NM_002556; amino acids 87–185 | For GFP tagged phospholipid probes |
| Gene (*H. sapiens*) | 2×ING2(PlantHD) | https://doi.org/10.1016/s0092-8674(03)00,480x | NM_001564; amino acids 190–280 | For GFP tagged phospholipid probes |
| Gene (*H. sapiens*) | 2×TAPP1(PHD) | https://doi.org/10.1042/bj3510019 | NM_001001974; amino acids 184–304 | For GFP tagged phospholipid probes |
| Gene (*H. sapiens*) | 2×TRPML1(PHD) | https://doi.org/10.1038/ncomms1037 | NM_020533; amino acids 1–69 | For GFP tagged phospholipid probes |
| Gene (*H. sapiens*) | Btk | https://doi.org/10.1016/s0969-2126(99)80057-4 | NM_000061; amino acids 1–177 | For GFP tagged phospholipid probes |
| Gene (*H. sapiens*) | PLCd1 | https://doi.org/10.1083/jcb.143.2.501 | NM_017035; amino acids 1–175 | For GFP tagged phospholipid probes |
| Gene (*S. cerevisiae*) | Sac1 | NCBI Reference Sequence | NM_001179777 | |
| Antibody | Mouse monoclonal anti-TOMM20 | Santa Cruz Biotechnology, Inc. | sc-11415 | 1:10,000 for WB |
| Antibody | Rabbit polyclonal anti-LAMP1 | Abcam | ab24170 | 1:10,000 for WB, 1:1000 for IF |
| Antibody | Rabbit polyclonal anti-p62 | MBL | PM045 | 1:10,000 for WB |
| Antibody | Rabbit polyclonal anti-LC3 | https://doi.org/10.1093/emboj/19.21.5720 | | 1:10,000 for WB |
| Antibody | HRP-conjugated anti-mouse IgG | Jackson ImmunoResearch Laboratories | 111-035-003 | 1:10,000 for WB |
| Antibody | HRP-conjugated anti-Rabbit IgG | Jackson ImmunoResearch Laboratories | 111-035-144 | 1:10,000 for WB |
| Antibody | Alexa Fluor 660-anti-rabbit IgG | Molecular Probes | A-21074 | 1:1000 for IF |
| Cell line (*H. sapiens*) | HeLa | RIKEN | RCB0007 | |

*Continued on next page*

*Continued*

| Reagent type (species) or resource | Designation | Source or reference | Identifiers | Additional information |
|---|---|---|---|---|
| Cell line (*H. sapiens*) | HEK293T | RIKEN | RCB2202 | |
| Cell line (*M. musculus*) | MEF | https://doi.org/10.1016/j.cell.2012.11.001 | | Established from C57BL/6 mice |
| Cell line (*H. sapiens*) | *STX17* KO HeLa | https://doi.org/10.1083/jcb.201712058 | | |
| Cell line (*H. sapiens*) | ATG8 hexa KO HeLa | https://doi.org/10.1083/jcb.201607039 | | Kindly provided by Michael Lazarou |
| Chemical compound, drug | Lipofectamine 2000 | Thermo Fisher Scientific | 11668019 | |
| Chemical compound, drug | FuGENE HD | Promega | VPE2311 | |
| Chemical compound, drug | Lysotracker Red DND99 | Thermo Fisher Scientific | L7528 | 50 nM |
| Chemical compound, drug | LysoTracker Deep Red | Thermo Fisher Scientific | L12492 | 50 nM |
| Chemical compound, drug | SaraFluor 650T HaloTag ligand | GoryoChemical | A308-02 | |
| Chemical compound, drug | Cellfectin II | Thermo Fisher Scientific | 10362100 | |
| Chemical compound, drug | Glutathione Sepharose 4B | GE Healthcare | 17075601 | |
| Chemical compound, drug | HRV3C protease | Fujifilm Wako Pure Chemical Corp. | 206–18151 | |
| Chemical compound, drug | CBB Stain One Super | Nacalai Tesque | 11642–31 | |
| Chemical compound, drug | DOPC | Avanti Polar Lipids | 850375 C | |
| Chemical compound, drug | DOPE | Avanti Polar Lipids | 850725 C | |
| Chemical compound, drug | DOPS | Avanti Polar Lipids | 840035 P | |
| Chemical compound, drug | 18:1 PI | Avanti Polar Lipids | 850149 P | |
| Chemical compound, drug | 18:1 PI3P | Avanti Polar Lipids | 850150 P | |
| Chemical compound, drug | 18:1 PI4P | Avanti Polar Lipids | 850151 P | |
| Chemical compound, drug | DSPE-PEG(2000) Biotin | Avanti Polar Lipids | 880129 C | |
| Chemical compound, drug | 18:1 Liss Phod PE | Avanti Polar Lipids | 810150 C | |
| Chemical compound, drug | OptiPrep | Cosmo Bio | 1893 | |
| Chemical compound, drug | NeutrAvidin Protein | Thermo Fisher Scientific | 31000 | |
| Chemical compound, drug | Lipofectamine RNAiMAX | Thermo Fisher Scientific | 13778150 | |

*Continued on next page*

*Continued*

| Reagent type (species) or resource | Designation | Source or reference | Identifiers | Additional information |
|---|---|---|---|---|
| Chemical compound, drug | digitonin | Sigma-Aldrich | D141 | |
| Chemical compound, drug | polybrane | Sigma-Aldrich | H9268 | |
| Chemical compound, drug | puromycin | Sigma-Aldrich | P8833 | |
| Chemical compound, drug | blasticidin | Fujifilm Wako Pure Chemical Corp. | 2218713 | |
| Chemical compound, drug | geneticin | Thermo Fisher Scientific | 10131 | |
| Chemical compound, drug | zeocin | Thermo Fisher Scientific | R25005 | |
| Commercial assay or kit | mMACHINE SP6 Transcription Kit | Thermo Fisher Scientific | AM1340 | |
| Commercial assay or kit | Rabbit reticulocyte lysates | Promega | L4960 | |
| Strain (*E. coli*) | DH10Bac | Thermo Fisher Scientific | 10361012 | |
| Cell line (*T. ni*) | High Five | Thermo Fisher Scientific | BTI-TN-5B1-4; B85502 | |
| Recombinant DNA reagent | *D. melanogaster* cDNA | Kindly provided by Masayuki Miura | | |
| Recombinant DNA reagent | *C. elegans* cDNA | Kindly provided by Hiroyuki Arai | | |
| Recombinant DNA reagent (plasmid) | pFastBac Dual Expression vector | Thermo Fisher Scientific | 10712024 | |
| Recombinant DNA reagent (plasmid) | GFP–Evectin-2 | Kindly provided by Hiroyuki Arai | | For GFP-tagged phospholipid probes |
| Recombinant DNA reagent (plasmid) | GFP–PKD C1ab | Kindly provided by Tamas Balla | | For GFP-tagged phospholipid probes |
| Recombinant DNA reagent (plasmid) | mRFP–2×FYVE | Kindly provided by Harald Stenmark | | For GFP-tagged phospholipid probes |
| Recombinant DNA reagent (plasmid) | GFP–P4M-SidMx2 | Addgene | 51472 | For GFP-tagged phospholipid probes |
| Recombinant DNA reagent (plasmid) | pCG-gag-pol | Kindly provided by Teruhiko Yasui | | For GFP-tagged phospholipid probes |
| Recombinant DNA reagent (plasmid) | pCG-VSV-G | Kindly provided by Teruhiko Yasui | | For GFP-tagged phospholipid probes |
| Recombinant DNA reagent (plasmid) | pMRXIP-GFP-STX17TM(DDDDD) | This paper | SN104 | *Figure 1* |
| Recombinant DNA reagent (plasmid) | pMRXIP-GFP-STX17TMΔC | This paper | SN106 | *Figure 1*, *Figure 1—figure supplement 1* |
| Recombinant DNA reagent (plasmid) | pMRXIP-GFP-STX17TM(RRRRR) | This paper | SN118 | *Figure 1*, *Figure 1—figure supplement 1* |
| Recombinant DNA reagent (plasmid) | pMRXIP-GFP-STX17TM(KKKKK) | This paper | SN84 | *Figure 1*, *Figure 1—figure supplement 1* |
| Recombinant DNA reagent (plasmid) | pMRXIP-GFP-STX17TM(AAAAA) | This paper | SN85 | *Figure 1*, *Figure 1—figure supplement 1* |

*Continued on next page*

*Continued*

| Reagent type (species) or resource | Designation | Source or reference | Identifiers | Additional information |
|---|---|---|---|---|
| Recombinant DNA reagent (plasmid) | pMRXIP-GFP-STX17TM(0KR) | This paper | SN178 | *Figure 1, Figure 1—figure supplement 1* |
| Recombinant DNA reagent (plasmid) | pMRXIP-GFP-STX17TM(1KR) | This paper | SN177 | *Figure 1, Figure 1—figure supplement 1* |
| Recombinant DNA reagent (plasmid) | pMRXIP-GFP-STX17TM(2KR) | This paper | SN168 | *Figure 1, Figure 1—figure supplement 1* |
| Recombinant DNA reagent (plasmid) | pMRXIP-GFP-STX17TM(3KR) | This paper | SN159 | *Figure 1, Figure 1—figure supplement 1* |
| Recombinant DNA reagent (plasmid) | pMRXIP-GFP-STX17TM | https://doi.org/10.1016/j.cell.2012.11.001 | Addgene; 45910 | *Figure 1, Figure 1—figure supplement 1, Figure 3—figure supplement 2* |
| Recombinant DNA reagent (plasmid) | pMRXIB-mRuby3-LC3 | This paper | SN219 | *Figure 1, Figure 1—figure supplement 1, Figure 2, Figure 2—figure supplement 1, Figure 3—figure supplement 1, Figure 3—figure supplement 2, Figure 4* |
| Recombinant DNA reagent (plasmid) | pMRXIP-GFP-Dmela STX17TM | This paper | SN162 | *Figure 1—figure supplement 1* |
| Recombinant DNA reagent (plasmid) | pMRXIP-GFP-Celegans Syx17TM | This paper | SN163 | *Figure 1—figure supplement 1* |
| Recombinant DNA reagent (plasmid) | pMRXIP-GFP-9K0Q | This paper | SN267 | *Figure 2, Figure 2—figure supplement 1* |
| Recombinant DNA reagent (plasmid) | pMRXIP-GFP-5K4Q | This paper | SN268 | *Figure 2, Figure 2—figure supplement 1* |
| Recombinant DNA reagent (plasmid) | pMRXIP-GFP-3K6Q | This paper | SN269 | *Figure 2, Figure 2—figure supplement 1* |
| Recombinant DNA reagent (plasmid) | pMRXIP-GFP-1K8Q | This paper | SN270 | *Figure 2, Figure 2—figure supplement 1* |
| Recombinant DNA reagent (plasmid) | pMRXIP-GFP-0K9Q | This paper | SN277 | *Figure 2, Figure 2—figure supplement 1* |
| Recombinant DNA reagent (plasmid) | pMRXIB-mRuby3-STX17TM | This paper | SN236 | *Figure 2, Figure 2—figure supplement 1, Figure 3, Figure 3—figure supplement 1, Figure 3—figure supplement 2, Figure 4—figure supplement 1* |
| Recombinant DNA reagent (plasmid) | pMRXIP-GFP-OSBP(PHD) | This paper | SN128 | *Figure 3, Figure 3—figure supplement 1* |
| Recombinant DNA reagent (plasmid) | pMRXIP-GFP-CERT(PHD) | This paper | SN232 | *Figure 3, Figure 3—figure supplements 1 and 2* |
| Recombinant DNA reagent (plasmid) | pMRXIP-P4M-SidMx2 | This paper | SN247 | *Figure 3, Figure 3—figure supplement 2* |
| Recombinant DNA reagent (plasmid) | pMRXIP-GFP-FAPP(PHD) | This paper | SN231 | *Figure 3, Figure 3—figure supplement 2* |
| Recombinant DNA reagent (plasmid) | pMRXIP-HaloTag7-LC3 | https://doi.org/10.7554/eLife.78923 | Addgene; 184899 | *Figure 3, Figure 4—figure supplement 1* |
| Recombinant DNA reagent (plasmid) | pMRXIB-WIPI2b-mRuby3 | This paper | SN214 | *Figure 3C* |
| Recombinant DNA reagent (plasmid) | pMRXIP-GFP-2xFYVE | This paper | SN262 | *Figure 3—figure supplement 1* |
| Recombinant DNA reagent (plasmid) | pMRXIP-GFP-ING2(PHD) | This paper | SN129 | *Figure 3—figure supplement 1* |

*Continued on next page*

*Continued*

| Reagent type (species) or resource | Designation | Source or reference | Identifiers | Additional information |
|---|---|---|---|---|
| Recombinant DNA reagent (plasmid) | pMRXIP-GFP-TRPML1(PHD) | This paper | SN132 | *Figure 3—figure supplement 1* |
| Recombinant DNA reagent (plasmid) | pMRXIP-GFP-PLCd1(PHD) | This paper | SN131 | *Figure 3—figure supplement 1* |
| Recombinant DNA reagent (plasmid) | pMRXIP-GFP-Evectin-2 | This paper | SN115 | *Figure 3—figure supplement 1* |
| Recombinant DNA reagent (plasmid) | pMRXIP-GFP-PKD C1ab | This paper | SN125 | *Figure 3—figure supplement 1* |
| Recombinant DNA reagent (plasmid) | pMRXIP-Btk1(PHD)-GFP | This paper | SN133 | *Figure 3—figure supplement 1A* |
| Recombinant DNA reagent (plasmid) | pMRXIP-GFP-TAPP1(PHD) | This paper | SN130 | *Figure 3—figure supplement 1A* |
| Recombinant DNA reagent (plasmid) | pMRXIP-GFP-Spo20(PABD) | This paper | SN124 | *Figure 3—figure supplement 1A* |
| Recombinant DNA reagent (plasmid) | pMRXIP-GFP-PI4KB | This paper | SN199 | *Figure 3—figure supplement 1C* |
| Recombinant DNA reagent (plasmid) | pMRXIP-GFP-PI4K2A | This paper | SN190 | *Figure 3—figure supplement 1C* |
| Recombinant DNA reagent (plasmid) | pMRXIP-GFP-CERT(PHD) (W33A) | This paper | pmSS123 | *Figure 3—figure supplement 2* |
| Recombinant DNA reagent (plasmid) | pMRXIB-mRuby3-CERT(PHD) | This paper | SN313 | *Figure 3—figure supplement 2, Figure 4—figure supplement 1* |
| Recombinant DNA reagent (plasmid) | pFastBacDual-GST-PreSci-ScSac1PD (WT) | This paper | HY580 | *Figure 4* |
| Recombinant DNA reagent (plasmid) | pFastBacDual-GST-PreSci-ScSac1PD (C392S) | This paper | HY581 | *Figure 4* |
| Recombinant DNA reagent (plasmid) | pFastBacDual-GST-PreSci-TEV-mGFP-STX17TM | This paper | HY1370 | *Figure 4* |
| Sequence-based reagent | human YKT6 siRNA antisense | https://doi.org/10.1083/jcb.201712058 | | GGTGTGGTCATTGCTGACAATGAAT |
| Sequence-based reagent | human YKT6 siRNA antisense sense | https://doi.org/10.1083/jcb.201712058 | | ATTCATTGTCAGCAATGACCACACC |
| Sequence-based reagent | human STX17 siRNA antisense | https://doi.org/10.1016/j.cell.2012.11.001 | | AATTAAGTCCGCTTCTAAGGTTTCC |
| Sequence-based reagent | human STX17 siRNA antisense sense | https://doi.org/10.1016/j.cell.2012.11.001 | | GGAAACCTTAGAAGCGGACTTAATT |
| Software, algorithm | FIJI-Image J | https://imagej.net/Fiji/Downloads | Image analysis were done using Fiji-Image J and plugins | |
| Software, algorithm | Illustrator | Adobe | Images were mounted using these softwares | |
| Software, algorithm | GraphPad prism | GraphPad Prism | Graphs and statistical tests were done using GraphPad Prism | |

## Plasmids and antibodies

First, cDNAs encoding human STX17 (*Itakura et al., 2012*), *D. melanogaster* STX17 (NM_079202), *C. elegans* STX17 (NM_059941), rat LC3B (*Hara et al., 2008*), and mouse ATG5 (NM_053069) were inserted into pMRXIP (harboring a puromycin-resistant marker) (*Kitamura et al., 2003*; *Ryckaert et al.,*

*1977*), pMRXIZ (harboring a zeocin-resistant marker) (*Morita et al., 2018*), and pMRXIB (harboring a blasticidin-resistant marker) (*Morita et al., 2018*) together with enhanced GFP or mRuby3 (codon-optimized, Addgene #74252). HaloTag7–LC3 was described previously (Addgene #184899) (*Yim et al., 2022*). STX17 fragments and their point mutations were generated by a standard PCR method or PCR-mediated site-directed mutagenesis.

Membrane surface charge probes were constructed by annealing with the following oligonucleotides (purchased from Thermo Fisher Scientific), after which they were inserted into pMRXIP plasmid with enhanced GFP:

GFP–9K0Q (GGCTCGGGGATCCGGGAATTCATCCAAAGATGGAAAAAAAAAGAAGAAGAA AAGTAAAACCAAATGCGTGATTATGTAACTCGAGAGCGGCCGCT, AGCGGCCGCTCTCGAG TTACATAATCACGCATTTGGTTTTACTTTTCTTCTTCTTTTTTTTTCCATCTTTGGATGAATTC CCGGATCCCGAGCC);
GFP–5K4Q (GGCTCGGGGATCCGGGAATTCAAGCAAAGACGGCCAGCAGCAACAAAAG AAGTCTAAGACCAAGTGTGTAATCATGTAACTCGAGAGCGGCCGCT, AGCGGCCGCTCT CGAGTTACATGATTACACACTTGGTCTTAGACTTCTTTTGTTGCTGCTGGCCGTCTTTGCTTGA ATTCCCGGATCCCGAGCC);
GFP–3K6Q (GGCTCGGGGATCCGGGAATTCATCCAAGGACGGACAACAGCAGCAACAACA GAGTAAAACTAAATGCGTGATAATGTAACTCGAGAGCGGCCGCT, AGCGGCCGCTCTCGAG TTACATTATCACGCATTTAGTTTTACTCTGTTGTTGCTGCTGTTGTCCGTCCTTGGATGAATTCC CGGATCCCGAGCC);
GFP–1K8Q (GGCTCGGGGATCCGGGAATTCATCCCAGGACGGTCAGCAGCAACAACAGCA ATCACAAACTAAATGTGTAATAATGTAACTCGAGAGCGGCCGCT, AGCGGCCGCTCTCGAG TTACATTATTACACATTTAGTTTGTGATTGCTGTTGTTGCTGCTGACCGTCCTGGGATGAATTC CCGGATCCCGAGCC).

GFP-0K9Q was generated by PCR-mediated site-directed mutagenesis.

To generate lipid probes, cDNAs encoding human 2×Spo20(PABD) (NM_005633; amino acids 422–551) (*Kassas et al., 2017*), CERT(PHD) (NM_001130105; amino acids 1–116) (*Hanada, 2006*), FAPP (NM_001807; amino acids 1–101) (*Andersen, 1983*), OSBP (NM_002556; amino acids 87–185) (*Levine and Munro, 1998*), 2×ING2(PlantHD) (NM_001564; amino acids 190–280) (*Gozani et al., 2003*), 2×TAPP1(PHD) (NM_001001974; amino acids 184–304) (*Dowler et al., 2000*), 2×TRPM-L1(PHD) (NM_020533; amino acids 1–69) (*Dong et al., 2010*), Btk (NM_000061; amino acids 1–177) (*Baraldi et al., 1999*), and rat PLCd1 (NM_017035; amino acids 1–175) (*Várnai and Balla, 1998*) were inserted into pMRXIP. GFP–Evectin-2 was provided by Hiroyuki Arai; GFP–PKD C1ab was provided by Tamas Balla; and mRFP–2×FYVE was provided by Harald Stenmark. GFP–P4M-SidMx2 (51472; Addgene) was amplified by PCR and subcloned into pMRXIP.

For the in vitro autophagosome recruitment assay, DNA fragments encoding GST–HRV3C-tagged yeast Sac1 (phosphatase domain; 2–517 amino acids) (NM_001179777) or its C392S mutant and mGFP (monomeric enhanced GFP with A206K mutation)–STX17TM were inserted downstream of the polyhedrin promoter of the pFastBac Dual Expression vector (10712024; Thermo Fisher Scientific).

For immunoblotting, mouse monoclonal anti-TOMM20 (sc-11415; Santa Cruz Biotechnology, Inc), rabbit polyclonal anti-LAMP1 (ab24170; Abcam), anti-p62 (PM045; MBL), and anti-LC3B (*Kabeya et al., 2000*) antibodies were used as primary antibodies. HRP-conjugated anti-mouse IgG and HRP-conjugated anti-rabbit IgG (111-035-003, 111-035-144; Jackson ImmunoResearch Laboratories) antibodies were used as secondary antibodies.

For immunostaining, rabbit polyclonal anti-LAMP1 (ab24170; Abcam) was used as a primary antibody, while Alexa Fluor 660-anti-rabbit IgG (A-21074; Molecular Probes) was used as secondary antibodies.

## Cell culture

Mouse embryonic fibroblasts (MEFs), U2OS cells, HeLa cells, and human embryonic kidney (HEK) 293T cells were cultured in Dulbecco's modified Eagle's medium (DMEM) (D6546; Sigma-Aldrich) supplemented with 10% fetal bovine serum (FBS) (S1820500; Biowest) and 2 mM L-glutamine (25030081; Gibco) in a 5% $CO_2$ incubator. HeLa, U2OS, and HEK293T cells were validated by STR profiling and authenticated by RIKEN. MEFs were established from C57BL/6 mice. The mammalian cell lines were

confirmed to be negative for mycoplasma contamination by observation by fluorescence microscopy. For the starvation treatment, cells were washed twice with phosphate-buffered saline (PBS) and incubated in amino acid-free DMEM (04833575; Fujifilm Wako Pure Chemical Corp.) without FBS. The *STX17* KO HeLa cells used in the present study were as previously described (*Matsui et al., 2018*). ATG8 hexa KO HeLa cells (lacking LC3A, LC3B, LC3C, GABARAP, GABARAPL1, and GABARAPL2) were kindly provided by Michael Lazarou (*Nguyen et al., 2016*).

For transient expression of mRuby3–CERT(PHD), HeLa cells were transiently transfected using FuGENE HD (VPE2311; Promega) with pMRXIB-mRuby3–CERT(PHD) for 24 hr in Opti-MEM (31985–070; Gibco).

## Retroviral infections and generation of stable cell lines

HEK293T cells were transiently transfected using Lipofectamine 2000 (11668019; Thermo Fisher Scientific) or FuGENE HD (VPE2311; Promega) with retrovirus vectors, pCG-gag-pol, and pCG-VSV-G (provided by Teruhiko Yasui). After cells were cultured for 2–3 days, the supernatant was collected and filtered through a 0.45 μm syringe filter unit (SLHV033RB; EMD Millipore). The obtained cells were cultured with retrovirus and 8 μg/ml polybrane (H9268; Sigma-Aldrich). Uninfected cells were eliminated by puromycin (P8833; Sigma-Aldrich), blasticidin (02218713; Fujifilm Wako Pure Chemical Corp.), geneticin (10131; Thermo Fisher Scientific), and zeocin (R25005; Thermo Fisher Scientific).

## Fluorescence microscopy

Live-cell fluorescence imaging was performed using a Delta Vision Elite widefield fluorescence microscope (GE Healthcare Life Science) equipped with a 100×PlanAPO oil-immersion objective lens (Olympus, NA1.40) and a cooled-CCD camera (Photometrics, CoolSNAP HQ2). Cells stably expressing GFP and mRuby3 were grown on a glass-bottom dish (617870; Greiner bio-one, or 11004006; IWAKI). To observe lysosomes, 50 nM Lysotracker Red DND99 (L7528; Thermo Fisher Scientific) or LysoTracker Deep Red (L12492; Thermo Fisher Scientific) was added to the medium. To observe HaloTag–LC3B, cells were observed in the presence of 200 nM SaraFluor 650T HaloTag ligand (A308-02; GoryoChemical). During live-cell imaging, the culture dish was mounted in a chamber (INUB-ONI-F2; TOKAI HIT) to maintain the culture conditions (37 °C, 5% $CO_2$). Images were acquired at 30 s intervals. Time series of 16-bit images were converted into RGB Tiff images using ImageJ software (Rasband, W.S., ImageJ, U. S. National Institutes of Health, http://imagej.nih.gov/ij/, 1997–2018). GFP intensity of autophagosomal structures was determined using ImageJ. Additionally, mRuby3–LC3 or Atg5-positive structures, including punctate and cup-shape structures (premature autophagosomes) and ring-like structures (mature autophagosomes) were extracted by the Analyze Particle function after binarization using the Auto Threshold v1.17 plugin (Method, Max Entropy). The GFP intensity of each particle was measured by the Measure ImageJ plugin, and the background signal (i.e. average GFP intensity of the entire cell) was subtracted.

The GFP–STX17TM intensity of mature autophagosomes or liposomes was determined using ImageJ. The center of ring-like structures in the red channel (LC3B or liposome signals) was first selected using the multi-point tool. The region of interest (ROI) was then defined by drawing a 10-μm-diameter circle around the previously determined center and capturing the red channel (LC3B or liposome signals). The GFP–STX17TM intensity was measured by the Radial Profile Plot ImageJ plugin, and the background signal of the surrounding cytosol area was subtracted.

## Preparation of recombinant proteins

In vitro transcription and translation in the rabbit reticulocyte lysate system were performed according to a published method with minor modifications (*Sharma et al., 2010*). For in vitro mRNA synthesis, DNA templates containing the SP6 RNA polymerase promoter site upstream of the sequence to be transcribed were generated by a standard PCR method and then incubated for 1 hr at 37 °C using the mMACHINE SP6 Transcription Kit (AM1340; Thermo Fisher Scientific). For in vitro protein synthesis, transcripts were incubated with rabbit reticulocyte lysates (L4960; Promega) for 30 min at 37 °C.

Recombinant yeast Sac1 (amino acids 2–517) and mGFP–STX17TM (amino acids 229–302) proteins were prepared using the Bac-to-Bac baculovirus expression system (10359016; Thermo Fisher Scientific). Bacmid DNAs were prepared using DH10Bac *E. coli* cells (10361012; Thermo Fisher Scientific), after which High Five insect cells (BTI-TN-5B1-4; B85502; Thermo Fisher Scientific) were transfected

with the bacmid DNAs using Cellfectin II (10362100; Thermo Fisher Scientific). Baculoviruses were generated and amplified in the High Five cells. After a 96 hr viral infection, the cells were suspended in 2 ml of ice-cold homogenization buffer (20 mM HEPES-KOH, pH 7.2, 250 mM sucrose, 1 mM EDTA, and protease inhibitor cocktail). Cells were then disrupted by ultrasonication using a UD-201 ultrasonic disruptor (TOMY). After centrifugation at 16,500×$g$ for 10 min, the supernatants were incubated with Glutathione Sepharose 4B (17075601; GE Healthcare) for 3 hr at 4 °C with gentle rotation. The sepharose resins were washed three times with homogenization buffer and then treated with HRV3C protease (206–18151; Fujifilm Wako Pure Chemical Corp.) for 12 hr at 16 °C. The eluted proteins were stored at −80 °C. To confirm recombinant expression, purified samples were separated by SDS-PAGE and detected by CBB staining using CBB Stain One Super (11642–31; Nacalai Tesque).

## Preparation of phospholipid vesicles

Liposomes were prepared as follows. First, 1,2-dioleoyl-sn-glycero-3-phosphocholine (DOPC) (850375 C; Avanti Polar Lipids), 1,2-dioleoyl-sn-glycero-3-phosphoethanolamine (DOPE) (850725 C; Avanti Polar Lipids), 1,2-dioleoyl-sn-glycero-3-phospho-L-serine (DOPS) (840035 P; Avanti Polar Lipids), 1,2-dioleoyl-sn-glycero-3-phospho-(1'-myo-inositol) (18:1 PI) (850149 P; Avanti Polar Lipids), 1,2-dioleoyl-sn-glycero-3-phospho-(1'-myo-inositol-3'-phosphate) (18:1 PI3P) (850150 P; Avanti Polar Lipids), and 1,2-dioleoyl-sn-glycero-3-phospho-(1'-myo-inositol-4'-phosphate) (18:1 PI4P) (850151 P; Avanti Polar Lipids) dissolved in chloroform to 1 µM final concentrations were mixed in a glass tube at the indicated ratios in the figure legends. Then, 1,2-distearoyl-sn-glycero-3-phosphoethanolamine-N-[biotinyl(polyethylene glycol)–2000] (DSPE-PEG(2000) Biotin) (880129 C; Avanti Polar Lipids) and 1,2-dioleoyl-sn-glycero-3-phosphoethanolamine-N-(lissamine rhodamine B sulfonyl) (18:1 Liss Phod PE) (810150 C; Avanti Polar Lipids) were added to the mixtures to label the liposomes. The chloroform was evaporated under argon gas and then completely dried in a vacuum desiccator overnight. The lipid film was hydrated in KHM buffer (20 mM HEPES-NaOH [pH 7.4], 110 mM potassium acetate, 2 mM MgCl$_2$) at a concentration of 1 mM and incubated for 16 hr at 30 °C.

## Purification of mature autophagosomes

*STX17* KO HeLa cells stably expressing mRuby3–LC3B were harvested from two 10 cm dishes and washed twice with ice-cold PBS. The cell pellets were collected after centrifugation at 700 × $g$ for 5 min and resuspended in 1 ml of ice-cold homogenization buffer (20 mM HEPES-KOH, pH7.2, 250 mM sucrose, 1 mM EDTA, and protease inhibitor cocktail). Cells were then disrupted by passage through a 25-gauge needle. The homogenized cells were centrifuged at 3000 × $g$ for 10 min to remove cell debris and undisrupted cells. The supernatant was diluted with an equal volume of 50% OptiPrep (1893; Cosmo Bio) with complete EDTA-free protease inhibitor (11873580001; Roche). Discontinuous OptiPrep gradients were generated in MLS-50 tubes (344057; Beckman Coulter) by overlaying each of the following OptiPrep solutions in homogenization buffer (20 mM HEPES-KOH [pH 7.4], 250 mM sucrose, 1 mM EDTA): 1.25 ml of the diluted supernatant in 25% OptiPrep, 0.25 ml in 20%, 0.75 ml in 15%, 0.75 ml in 10%, 2.0 ml in 5%, and 0.25 ml in 0%. The gradients were centrifuged at 150,000×$g$ in MLS-50 rotors (Beckman Instruments) for 3 hr, and subsequently, 10 fractions (0.5 ml each) were collected from the top. Proteins in each fraction were isolated by TCA precipitation. The samples were separated by SDS-PAGE and transferred to Immobilon-P polyvinylidene difluoride membranes (IPVH00010; EMD Millipore). Immunoblotting analysis was performed with the indicated antibodies. Immobilon Western Chemiluminescent HRP Substrate (P90715; EMD Millipore) was used to visualize the signals, which were detected on an IQ800 biomolecular imager (Cytiva). Contrast and brightness adjustments were performed using Photoshop 2022 (Adobe).

## In vitro recruitment assay

For the liposome binding assay, the prepared liposomes were mixed with recombinant proteins produced by rabbit reticulocyte lysate for 15 min at 30 °C. A 0.2-mm-thick chamber was created by attaching two glass cover slips (Matunami glass) together with double-sided tape applied along the long edges. NeutrAvidin Protein (31000; Thermo Fisher Scientific) was added to the chamber for 15 min at room temperature before being washed out with KHM buffer. The liposome–recombinant protein mixture was then transferred to the NeutrAvidin Protein-coated chamber. Images

were captured immediately after using a confocal microscope (FV3000; Olympus) equipped with a 60×PlanAPO oil-immersion objective lens (Olympus, NA1.42).

For the autophagosome recruitment assay, the LC3-positive, LAMP1-negative autophagosomal fraction was incubated with 0.5 μg of Sac1 proteins from insect cells for 30 min at 37 °C. Then, mGFP–STX17TM was added to the mixture, which was then incubated again for 30 min at 37 °C. The mixtures were transferred to a glass-bottom dish (617870; Greiner bio-one), and images were captured using a fluorescence microscope (BZ-810; Keyence) equipped with a 60×oil-immersion objective lens (Nikon, NA1.40).

## RNA interference

Stealth RNAi oligonucleotides were purchased from Thermo Fisher Scientific. The following sequences were used: human YKT6 siRNA antisense, 5′- GGTGTGGTCATTGCTGACAATGAAT –3′; human YKT6 siRNA antisense sense, 5′- ATTCATTGTCAGCAATGACCACACC –3′ (*Matsui et al., 2018*). The siRNA oligonucleotides for human STX17 (HSS123732 antisense, 5′- AATTAAGTCCGCTTCTAAGGTTTC C –3′; HSS123732 sense, 5′- GGAAACCTTAGAAGCGGACTTAATT –3′) were previously reported (*Itakura et al., 2012*). The stealth RNAi oligonucleotides were transfected into cells using Lipofect-amine RNAiMAX (13778150; Thermo Fisher Scientific) according to the manufacturer's instructions. Three days after transfection, cells were used for analysis.

## Immunostaining

Cells grown on coverslips were washed with PBS and fixed with 4% paraformaldehyde phosphate buffer solution (0915485; Nacalai Tesque) for 10 min at room temperature. Fixed cells were permeabilized with 50 μg/ml digitonin (D141; Sigma-Aldrich) in PBS for 5 min, blocked with 3% BSA in PBS for 10 min, and then incubated with primary antibodies for 16 hr at 4 °C. After being washed three times with PBS, cells were incubated with secondary antibodies for 1 hr at room temperature. The coverslips were observed using a confocal laser microscope (FV3000; Olympus) equipped with a 60×PlanAPO oil-immersion objective lens (Olympus, NA1.42).

## Sequence alignment

Amino acid sequences of STX17 from the following species were obtained from the NCBI protein database: *Homo sapiens* (NP_060389.2), *Mus musculus* (NP_080619.2), *Danio rerio* (NP_001007450.1), *Ciona intestinalis* (NP_492342.1), *Drosophila melanogaster* (NP_523926.1), and *Caenorhabditis elegans* (NP_492342.1). The sequences were aligned using clustal W (https://www.genome.jp/tools-bin/clustalw) as implemented in Molecular Evolutionary Genetics Analysis X (*Kumar et al., 2018b*).

## Molecular dynamics simulation

We used the all-atom model for SXT17TM utilizing initial structures predicted by trRosetta, a method based on deep learning (*Du et al., 2021*). Because five different three-dimensional structural models were predicted (TM score >0.7), each was used in the simulations. The lipid bilayer was modeled using the HMMM model, which was developed to study protein–membrane interactions more efficiently by replacing the membrane lipids with short-chain lipid and organic solvents to facilitate lateral diffusion (*Ohkubo et al., 2012*). The lipid headgroups in the HMMM model were identical to those in the all-atom model and faithfully represented the membrane surface. The lipid membrane consisted of 150 lipid molecules in each leaflet. The membranes with and without PI4P were examined for lipid compositions of POPC:POPE:POPI4P (POPI14)=70:20:10, 70:30:0 or POPC:POPE:POPI = 70:20:10, respectively. The initial configuration was prepared so that the center of mass of STX17TM was located 3 nm above the membrane surface and the first principal axis of the atomic configuration was tilted 45 degrees from the z axis. The protein and lipids were solvated in a 10 nm × 10 nm×30 nm box with the TIP3P water model and 0.15 M KCl ions. The initial configurations were built by the Membrane Builder module in the CHARMM-GUI server (*Jo et al., 2008*). All molecular dynamics simulations were performed using GENESIS (*Jung et al., 2015*). The CHARMM36 force-field was used to describe the interactions in the system (*Huang et al., 2017*). Energy minimization was performed for 1000 steps by the steepest descent algorithm and then by the conjugate gradient algorithm. Then, a 250 ps NVT simulation was performed at 303.15 K for solvent equilibration, followed by a 1.6 ns NPT equilibration to 1 atm using the Langevin thermostat/barostat (*Quigley and Probert, 2004*). The

production simulations were performed for 100 ns with a time-step of 2.5 fs and the Langevin thermostat/barostat. Long-range electrostatic interactions were simulated using the particle-mesh Ewald method (*Darden et al., 1993*; *Essmann et al., 1995*). The short-range electrostatic and van der Waals interactions both used a cutoff of 12 Å. All bonds were constrained by the SHAKE/RATTLE algorithm (*Andersen, 1983*; *Ryckaert et al., 1977*).

## Statistical analysis

Comparisons between data from two groups were evaluated using Welch's *t*-test, and comparisons of data among multiple groups were performed using one-way analysis of variance (ANOVA) followed by the Dunnett's, Sidak's, or Tukey's multiple comparison tests as implemented in GraphPad Prism 8 and 9 (GraphPad Software). Data distributions were assumed to be normal, but this was not formally tested.

## Acknowledgements

We thank Shoji Yamaoka for providing pMRX-IP vector, Teruhito Yasui for the pCG-gag-pol and pCG-VSV-G vectors, Hiroyuki Arai for *C. elegans* cDNA and GFP–Evectin-2, Masayuki Miura for *D. melanogaster* cDNA, Tamas Balla for GFP–PKD C1ab, Harald Stenmark for mRFP–2xFYVE, Michael Lazarou for ATG8 hexa KO HeLa cells, Yuji Sugita and Ai Niitsu for helpful discussions on the molecular dynamics calculations, Koki Kamiya and Shoji Takeuchi for helping the liposome experiments, and Gábor Juhász for helpful discussion and sharing his unpublished results. The numerical computations were performed with the RIKEN supercomputer system (HOKUSAI). We thank Eisuke Itakura for technical advice on the in vitro experiments. We thank the members of the Mizushima Laboratory for helpful discussions, especially Maya Shirakawa for technical assistance with the experiments. This work was supported by Grants-in-Aid for Specially Promoted Research (22H04919 to NM), for Transformative Research Areas (A) (21H05256 to HY), for Scientific Research (C) (23K05715 to YS), and for JSPS Fellows (17J02747 to SS) from the Japan Society for the Promotion of Science (JSPS) and by the Exploratory Research for Advanced Technology (ERATO) research funding program of the Japan Science and Technology Agency (JST) (JPMJER1702 to NM). The numerical computations were performed with the supercomputer systems at RIKEN (HOKUSAI, Fugaku) and the University of Tokyo (Wisteria/BDEC-01).

## Additional information

### Competing interests

Noboru Mizushima: Reviewing editor, *eLife*. The other authors declare that no competing interests exist.

### Funding

| Funder | Grant reference number | Author |
| --- | --- | --- |
| Japan Society for the Promotion of Science | 22H04919 | Noboru Mizushima |
| Japan Society for the Promotion of Science | 21H05256 | Hayashi Yamamoto |
| Japan Society for the Promotion of Science | 23K05715 | Yuji Sakai |
| Japan Society for the Promotion of Science | 17J02747 | Saori Shinoda |
| Exploratory Research for Advanced Technology | JPMJER1702 | Noboru Mizushima |

The funders had no role in study design, data collection and interpretation, or the decision to submit the work for publication.

## Author contributions
Saori Shinoda, Conceptualization, Funding acquisition, Investigation, Writing – original draft, Writing – review and editing; Yuji Sakai, Formal analysis, Funding acquisition, Investigation, Writing – original draft, Writing – review and editing; Takahide Matsui, Masaaki Uematsu, Ikuko Koyama-Honda, Jun-ichi Sakamaki, Investigation, Writing – review and editing; Hayashi Yamamoto, Conceptualization, Supervision, Funding acquisition, Investigation, Writing – original draft, Writing – review and editing; Noboru Mizushima, Conceptualization, Supervision, Funding acquisition, Investigation, Writing – original draft, Project administration, Writing – review and editing

## Author ORCIDs
Saori Shinoda https://orcid.org/0009-0007-7366-7147
Yuji Sakai http://orcid.org/0000-0002-2184-4304
Takahide Matsui http://orcid.org/0000-0002-3342-0170
Masaaki Uematsu https://orcid.org/0000-0002-0197-8401
Ikuko Koyama-Honda http://orcid.org/0000-0001-9321-9682
Hayashi Yamamoto http://orcid.org/0000-0002-2831-1463
Noboru Mizushima http://orcid.org/0000-0002-6258-6444

Reviewer #1 (Public Review): https://doi.org/10.7554/eLife.92189.3.sa1
Reviewer #2 (Public Review): https://doi.org/10.7554/eLife.92189.3.sa2
Reviewer #3 (Public Review): https://doi.org/10.7554/eLife.92189.3.sa3
Author response https://doi.org/10.7554/eLife.92189.3.sa4

---

## Additional files

### Supplementary files
• MDAR checklist

### Data availability
All data generated or analysed during this study are included in the manuscript and supporting files.

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
