## [Editor Report · eLife assessment]

This paper addresses a **fundamental** issue in the field of autophagy: how is a protein responsible for autophagosome-lysosome fusion recruited to mature autophagosomes but not immature ones? The work succeeds in its ambition to provide a new conceptual advance. The evidence supporting the conclusions is **convincing**, with fluorescence microscopy, biochemical assays, and molecular dynamics simulations. This work will be of broad interest to cell biologists and biochemists studying autophagy, and also those focusing on lipid/membrane biology.

---

## [Referee Report · Reviewer #1 (Public Review)]

In this manuscript, the authors report a molecular mechanism for recruiting syntaxin 17 (Syn17) to the closed autophagosomes through the charge interaction between enriched PI4P and the C-terminal region of Syn17. How to precisely control the location and conformation of proteins is critical for maintaining autophagic flux. Particularly, the recruitment of Syn17 to autophagosomes remains unclear. In this paper, the author describes a simple lipid-protein interaction model beyond previous studies focusing on protein-protein interactions. This represents conceptual advances.

---

## [Referee Report · Reviewer #2 (Public Review)]

Summary:

Syntaxin17 (STX17) is a SNARE protein that is recruited to mature (i.e., closed) autophagosomes, but not to immature (i.e., unclosed) ones, and mediates the autophagosome-lysosome fusion. How STX17 recognizes the mature autophagosome is an unresolved interesting question in the autophagy field. Shinoda and colleagues set out to answer this question by focusing on the C-terminal domain of STX17 and found that PI4P is a strong candidate that causes the STX17 recruitment to the autophagosome.

Strengths:

The main findings are: (1) Rich positive charges in the C-terminal domain of STX17 are sufficient for the recruitment to the mature autophagosome; (2) Fluorescence charge sensors of different strengths suggest that autophagic membranes have negative charges and the charge increases as they mature; (3) Among a battery of fluorescence biosensors, only PI4P-binding biosensors distribute to the mature autophagosome; (4) STX17 bound to isolated autophagosomes is released by treatment with Sac1 phosphatase; (5) By dynamic molecular simulation, STX17 TM is shown to be inserted to a membrane containing PI4P but not to a membrane without it. These results indicate that PI4P is a strong candidate that STX17 binds to in the autophagosome.

Weaknesses:

• It was not answered whether PI4P is crucial for the STX17 recruitment in cells because manipulation of the PI4P content in autophagic membranes was not successful for unknown reasons.

• The question that the authors posed in the beginning, i.e., why is STX17 recruited to the mature (closed) autophagosome but not to immature autophagic membranes, was not answered. The authors speculate that the seemingly gradual increase of negative charges in autophagic membranes is caused by an increase in PI4P. However, this was not supported by the PI4P fluorescence biosensor experiment that showed their distribution to the mature autophagosome only. Here, there are at least two possibilities: (1) The increase of negative charges in immature autophagic membranes is derived from PI4P. However the fluorescence biosensors do not bind there for some reason; for example, they are not sensitive enough to recognize PI4P until it reaches a certain level, or simply, their binding does not occur in a quantitative manner. (2) The negative charge in immature membranes is not derived from PI4P, and PI4P is generated abundantly only after autophagosomes are closed. In either case, it is not easy to explain why STX17 is recruited to the mature autophagosome only. For the first scenario, it is not clear how the PI4P synthesis is regulated so that it reaches a sufficient level only after the membrane closure. In the second case, the mechanism that produces PI4P only after the autophagosome closure needs to be elucidated (so, in this case, the question of the temporal regulation issue remains the same).

---

## [Referee Report · Reviewer #3 (Public Review)]

Summary:

In this study, the authors set out to address the question of how the SNARE protein Syntaxin 17 senses autophagosome maturation by being recruited to autophagosomal membranes only once autophagosome formation and sealing is complete. The authors discover that the C-terminal region of Syntaxin 17 is essential for its sensing mechanism that involves two transmembrane domains and a positively charged region. The authors discover that the lipid PI4P is highly enriched in mature autophagosomes and that electrostatic interaction with Syntaxin 17's positively charged region with PI4P drives recruitment specifically to mature autophagosomes. The temporal basis for PI4P enrichment and Syntaxin 17 recruitment to ensure that unsealed autophagosomes do not fuse with lysosomes is a very interesting and important discovery. Overall, the data are clear and convincing, with the study providing important mechanistic insights that will be of broad interest to the autophagy field, and also to cell biologists interested in phosphoinositide lipid biology. The author's discovery also provides an opportunity for future research in which Syntaxin 17's c-terminal region could be used to target factors of interest to mature autophagosomes.

Strengths:

The study combines clear and convincing cell biology data with in vitro approaches to show how Syntaxin 17 is recruited to mature autophagosomes. The authors take a methodical approach to narrow down the critical regions within Syntaxin 17 required for recruitment and use a variety of biosensors to show that PI4P is enriched on mature autophagosomes.

Weaknesses:

There are no major weaknesses, overall the work is highly convincing. It would have been beneficial if the authors could have shown whether altering PI4P levels would affect Syntaxin 17 recruitment. However, this is understandably a challenging experiment to undertake and the authors outlined their various attempts to tackle this question.

---

## [Author Response]

The following is the authors’ response to the original reviews.

**Reviewer #1 (Public Review):**
In this manuscript, the authors report a molecular mechanism for recruiting syntaixn 17 (Syn17) to the closed autophagosomes through the charge interaction between enriched PI4P and the C-terminal region of Syn17. How to precisely control the location and conformation of proteins is critical for maintaining autophagic flux. Particularly, the recruitment of Syn17 to autophagosomes remains unclear. In this paper, the author describes a simple lipid-protein interaction model beyond previous studies focusing on protein-protein interactions. This represents conceptual advances.

We would like to thank Reviewer #1 for the positive evaluation of our study.

**Reviewer #2 (Public Review):**
Summary:Syntaxin17 (STX17) is a SNARE protein that is recruited to mature (i.e., closed) autophagosomes, but not to immature (i.e., unclosed) ones, and mediates the autophagosome-lysosome fusion. How STX17 recognizes the mature autophagosome is an unresolved interesting question in the autophagy field. Shinoda and colleagues set out to answer this question by focusing on the C-terminal domain of STX17 and found that PI4P is a strong candidate that causes the STX17 recruitment to the autophasome.Strengths:The main findings are: (1) Rich positive charges in the C-terminal domain of STX17 are sufficient for the recruitment to the mature autophagosome; (2) Fluorescence charge sensors of different strengths suggest that autophagic membranes have negative charges and the charge increases as they mature; (3) Among a battery of fluorescence biosensors, only PI4P-binding biosensors distribute to the mature autophagosome; (4) STX17 bound to isolated autophagosomes is released by treatment with Sac1 phosphatase; (5) By dynamic molecular simulation, STX17 TM is shown to be inserted to a membrane containing PI4P but not to a membrane without it. These results indicate that PI4P is a strong candidate that STX17 binds to in the autophagosome.

We would like to thank Reviewer #2 for pointing out these strengths.

Weaknesses:• It was not answered whether PI4P is crucial for the STX17 recruitment in cells because manipulation of the PI4P content in autophagic membranes was not successful for unknown reasons.

As we explained in the initial submission, we tried to deplete PI4P in autophagosomes by multiple methods but did not succeed. In this revised manuscript, we added the result of an experiment using the PI 4-kinase inhibitor NC03 (Figure 4―figure supplement 1), which shows no significant effect on the autophagosomal PI4P level and STX17 recruitment.

**Author response image 1. sa4fig1:** The PI 4-kinase inhibitor NC03 failed to suppress autophagosomal PI4P accumulation and STX17 recruitment. HEK293T cells stably expressing mRuby3–STX17TM (A) or mRuby3–CERT(PHD) (B) and Halotag-LC3 were cultured in starvation medium for 1 h and then treated with and without 10 μM NC03 for 10 min. Representative confocal images are shown. STX17TM- or CERT(PHD)-positive rates of LC3 structures per cell (n > 30 cells) are shown in the graphs. Solid horizontal lines indicate medians, boxes indicate the interquartile ranges (25th to 75th percentiles), and whiskers indicate the 5th to 95th percentiles. Differences were statistically analyzed by Welch’s t-test. Scale bars, 10 μm (main), 1 μm (inset).

• The molecular simulation study did not show whether PI4P is necessary for the STX17 TM insertion or whether other negatively charged lipids can play a similar role.

As the reviewer suggested, we performed the molecular dynamics simulation using membranes with phosphatidylinositol, a negatively charged lipid. STX17 TM approached the PI-containing membrane but was not inserted into the membrane within a time scale of 100 ns in simulations of all five structures. This data suggests that PI4P, which is more negatively charged than PI, is required for STX17 insertion. Thus, we have included these data in Figure 5E and F and added the following text to Lines 242–244.“Moreover, if the membrane contained phosphatidylinositol (PI) instead of PI4P, STX17 approached the PI-containing membrane but was not inserted into the membrane (Figure 5E, F, Video 3)."

**Author response image 2. sa4fig2:** Molecular dynamics simulations showing that phosphatidylinositol is not sufficient for STX17TM insertion into membranes. (E) An example of a time series of simulated results of STX17TM insertion into a membrane consisting of 70% phosphatidylcholine (PC), 20% phosphatidylethanolamine (PE), and 10% phosphatidylinositol (PI). STX17TM is shown in blue. Phosphorus in PC, PE and PI are indicated by yellow, cyan, and orange, respectively. Short-tailed lipids are represented as green sticks. The time evolution series are shown in Video 3. (F) Time evolution of the z-coordinate of the center of mass (z_cm) of the transmembrane helices of STX17TM in the case of membranes with PI. Five independent simulation results are represented by solid lines of different colors. The gray dashed lines indicate the locations of the lipid heads. A scale bar indicates 5 nm.

• The question that the authors posed in the beginning, i.e., why is STX17 recruited to the mature (closed) autophagosome but not to immature autophagic membranes, was not answered. The authors speculate that the seemingly gradual increase of negative charges in autophagic membranes is caused by an increase in PI4P. However, this was not supported by the PI4P fluorescence biosensor experiment that showed their distribution to the mature autophagosome only. Here, there are at least two possibilities: (1) The increase of negative charges in immature autophagic membranes is derived from PI4P. However the fluorescence biosensors do not bind there for some reason; for example, they are not sensitive enough to recognize PI4P until it reaches a certain level, or simply, their binding does not occur in a quantitative manner. (2) The negative charge in immature membranes is not derived from PI4P, and PI4P is generated abundantly only after autophagosomes are closed. In either case, it is not easy to explain why STX17 is recruited to the mature autophagosome only. For the first scenario, it is not clear how the PI4P synthesis is regulated so that it reaches a sufficient level only after the membrane closure. In the second case, the mechanism that produces PI4P only after the autophagosome closure needs to be elucidated (so, in this case, the question of the temporal regulation issue remains the same).

We thank the reviewers for pointing this out. While the probe for weakly negative charges (1K8Q) labeled both immature and mature autophagosomes, the probes for intermediate charges (5K4Q and 3K6Q) and PI4P labeled only mature autophagosomes (Figure 2F, Figure 2–figure supplement 1B). Thus, we think that the autophagosomal membrane rapidly and drastically becomes negatively charged, and at the same time, PI4P is enriched. Although immature membranes may have weak negative charges, we did not examine which lipids contribute to the negative charges. Thus, we have added the following sentences to the Discussion part.

“Our data of the 1K8Q probe suggest that immature autophagosomal membranes may also have slight negative charges (Figure 2E). Although the source of the negative charge of immature autophagosomes is currently unknown, it may be derived from low levels of PI4P, which is undetectable by the PI4P probes and/or other negatively charged lipids such as PI and PS (Schmitt et al., EMBO Rep, 2022).” (Lines 279–283)“In any case, it would be important to elucidate how PI 4-kinase activity or PI4P synthesis is upregulated during autophagosome maturation.” (Lines 302–303)

**Reviewer #3 (Public Review):**
Summary:In this study, the authors set out to address the question of how the SNARE protein Syntaxin 17 senses autophagosome maturation by being recruited to autophagosomal membranes only once autophagosome formation and sealing is complete. The authors discover that the C-terminal region of Syntaxin 17 is essential for its sensing mechanism that involves two transmembrane domains and a positively charged region. The authors discover that the lipid PI4P is highly enriched in mature autophagosomes and that electrostatic interaction with Syntaxin 17's positively charged region with PI4P drives recruitment specifically to mature autophagosomes. The temporal basis for PI4P enrichment and Syntaxin 17 recruitment to ensure that unsealed autophagosomes do not fuse with lysosomes is a very interesting and important discovery. Overall, the data are clear and convincing, with the study providing important mechanistic insights that will be of broad interest to the autophagy field, and also to cell biologists interested in phosphoinositide lipid biology. The author's discovery also provides an opportunity for future research in which Syntaxin 17's c-terminal region could be used to target factors of interest to mature autophagosomes.Strengths:The study combines clear and convincing cell biology data with in vitro approaches to show how Syntaxin 17 is recruited to mature autophagosomes. The authors take a methodical approach to narrow down the critical regions within Syntaxin 17 required for recruitment and use a variety of biosensors to show that PI4P is enriched on mature autophagosomes.

We would like to thank Reviewer #3 for the positive comments.

Weaknesses:There are no major weaknesses, overall the work is highly convincing. It would have been beneficial if the authors could have shown whether altering PI4P levels would affect Syntaxin 17 recruitment. However, this is understandably a challenging experiment to undertake and the authors outlined their various attempts to tackle this question.

We thank Reviewer #3 for pointing this out. Please see our above response to Reviewer #2 (Public Review).

In addition, clear statements within the figure legends on the number of independent experimental repeats that were conducted for experiments that were quantitated are not currently present in the manuscript.

As pointed out by Reviewer #3, we have added the number of independent experimental repeats in the figure legends.

**Reviewer #1 (Recommendations For The Authors):**
This paper is well written and all experiments were conducted with a high standard. Several minor issues should be addressed before final publication.(1) To further confirm the charge interaction, a charge screening experiment should be performed for Fig. 2A.

We have asked Reviewer #1 through the editor what this experiment meant and understood that it was to see the effects of high salt concentrations. We monitored the association of GFP-STX17TM with liposomes in the presence or absence of 1 M NaCl and found that it was blocked in a high ionic buffer. This data supports the electrostatic interaction of STX17 with membranes. We have included this data in Figure 2B and added the following sentences to Lines 124–126.

“The association of STX17TM with PI4P-containing membranes was abolished in the presence of 1 M NaCl (Figure 2B). These data suggest that STX17 can be recruited to negatively charged membranes via electrostatic interaction independent of the specific lipid species.”

**Author response image 3. sa4fig3:** GFP–STX17TM translated in vitro was incubated with rhodamine-labeled liposomes containing 70% PC, 20% PE and 10% PI4P in the presence of 1 M NaCl or 1. 2 M sucrose. GFP intensities of liposomes were quantified and shown as in Figure 1C (n > 30).

(2) The authors claim that "Autophagosomes become negatively charged during maturation", based on experiments using membrane charge probes. Since it's mainly about the membrane, it's better to refine the claim to "The membrane of autophasosomes becomes...", which would be more precise and close to the topic of this paper.

We would like to thank the reviewer for pointing this out. This point is valid. As recommended, we have collected the phrases “Autophagosomes become negatively charged during maturation” to “The membrane of autophagosomes becomes negatively charged during maturation” (Line 72, 118, 262, 969 (title of Figure2), 1068 (title of Figure2–figure supplyment1)).

(3) The authors should add more discussion regarding the "specificity" for recruiting Syn17 through the charge interaction. Particularly, how Syn17 could be maintained before the closure of autophagosomes? For the MD simulations in Fig. 5, the current results don't add much to the manuscript. The cell biology experiments have demonstrated the conclusion. The authors could try to find more details about the insertion by analyzing the simulation movies. Do membrane packing defects play a role during the insertion process? A similar analysis was conducted for alpha-synuclein (https://pubmed.ncbi.nlm.nih.gov/33437978/).

Regarding the mechanism of STX17 maintenance in the cytosol, we do not think that other molecules, such as chaperones, are essential because purified recombinant mGFP-STX17TM used in this study is soluble. However, it does not rule out such a mechanism, which would be a future study.

In the paper by Liu et al. (PMID: 33437978), small liposomes with diameters of 25–50 nm are used. Therefore, there are packing defects in the highly curved membranes, to which alpha-synuclein helices are inserted in a curvature-dependent manner. On the other hand, autophagosomes are much larger (~1 um in diameter) and almost flat for STX17 molecules, so we think it is unlikely that STX17 recognizes the packing defect.

**Reviewer #2 (Recommendations For The Authors):**
• The two (and other) possibilities with regards to the interpretation of the negative charge/PI4P result in autophagic membranes are hoped to be discussed.

As mentioned above, we have added the following sentences to the Discussion section.“Our data of the 1K8Q probe suggest that immature autophagosomal membranes may also have slight negative charges (Figure 2E). Although the source of the negative charge of immature autophagosomes is currently unknown, it may be derived from low levels of PI4P, which is undetectable by the PI4P probes and/or other negatively charged lipids such as PI and PS (Schmitt et al., EMBO Rep, 2022).” (Lines 279–283)

“In any case, it would be important to elucidate how PI 4-kinase activity or PI4P synthesis is upregulated during autophagosome maturation.” (Lines 302–303)

• Fluorescence biosensors are convenient to give an overview of the intracellular distribution of various lipids, but some of them show false-negative results. For example, evectin-2-PH for PS binds to endosomes but not to the plasma membrane, even though the latter contains abundant PS. With regards to PI4P, some biosensors illuminate both the Golgi and autophagosome, while others do not appear to bind the Golgi. Moreover, fluorescence biosensors for PI(3,5)P2 and PI(3,4)P2, which are also candidates for the STX17 insertion issue, are less reliable than others (e.g., those for PI3P and PI(4,5)P2). These problems need to be considered.

We agree with Reviewer #2 that fluorescence biosensors are not perfect for detecting specific lipids. Based on the Reviewer’s suggestion, we have included a comment on this in the Discussion section as follows (Lines 265–268).

“Given the possibility that fluorescence lipid probes may give false-negative results, a more comprehensive biochemical analysis, such as lipidomics analysis of mature autophagosomes, would be imperative to elucidate the potential involvement of other negatively charged lipids.”

• A negative control for the PI4P biosensor, i.e., a mutant lacking the PI4P binding ability, is better to be tested to confirm the presence of PI4P in autophagosomes.

We would like to thank the Reviewer for this comment. We conducted the suggested experiment and confirmed that the CERT(PHD)(W33A) mutant, which is deficient for PI4P binding (Sugiki et al., JBC. 2012), was diffusely present in the cytosol and did not localize to STX17-positive autophagosomes. This data supports our conclusion that PI4P is indeed present in autophagosomes. We have included this data in Figure 3–figure supplement 2A and explained it in the text (Lines 164–166).

**Author response image 4. sa4fig4:** Mouse embryonic fibroblasts (MEFs) stably expressing GFP–CERT(PHD)(W33A) and mRuby3–STX17TM were cultured in starvation medium for 1 h. Bars indicate 10 μm (main images) and 1 μm (insets).

• As a control to the molecular dynamic simulation study, STX17 TM insertion into a membrane containing other negative charge lipids, especially PI, needs to be tested. PI is a negative charge lipid that is likely to exist in autophagic membranes (as suggested by the authors' past study).

We thank the reviewers for this suggestion. As mentioned above (Reviewer #2, Public Review), we performed the molecular dynamics simulation using membranes containing PI and added the results in Figure 5E and F and Video 3.

• If the putative role of PI4P could be shown in the cellular context, the authors' conclusion would be much strengthened. I wonder if overexpression of PI4P fluorescence biosensors, especially those that appear to bind to the autophagosome almost exclusively, may suppress the recruitment of STX17 there.

We would like to thank the Reviewer for asking this question. In MEFs stably overexpressing PI4P probes driven by the CMV promoter, STX17 recruitment was not affected. Thus, simple overexpression of PI4P probes does not appear to be effective in masking PI4P in autophagosomes.

Another idea is to use an appropriate molecule (e.g., WIPI2, ATG5) and to recruit Sac1 to autophagic membranes by using the FRB-FKBP system or the like. I hope these and other possibilities will be tested to confirm the importance of PI4P in the temporal regulation of STX17 recruitment.

We tried the FRB-FKBP system using the phosphatase domain of yeast Sac1 fused to FKBP and LC3 fused to FRB, but unfortunately, this system failed to deplete PI4P from the autophagosomal membrane.

**Reviewer #3 (Recommendations For The Authors):**
A few areas for suggested improvement are:(1) It would be helpful if the authors could clarify for all figures how many independent experiments were conducted for all experiments, particularly those that have quantitation and statistical analyses.

As pointed out by Reviewer #3, we have added the number of independent experimental repeats in the figure legends.

The authors made several attempts to modulate PI4P levels on autophagosomes although understandably this proved to be challenging. A couple of suggestions are provided to address this area:(2) Given the reported role of GABARAPs in PI4K2a recruitment and PI4P production on autophagosomes, as well as autophagosome-lysosome fusion (Nguyen et al (2016) J Cell Biol) it would be worthwhile to assess whether GABARAP TKO cells have reduced PI4P and reduced Stx17 recruitment

According to the Reviewer’s suggestion, we examined the localization of STX17 TM and the PI4P probe CERT(PHD) in ATG8 family (LC3/GABARAP) hexa KO HeLa cells that were established by the Lazarou lab (Nguyen et al., JCB 2016). As in WT cells, STX17 TM and CERT(PHD) were still colocalized with each other in hexa KO cells, suggesting that neither STX17 recruitment nor PI4P enrichment depends on ATG8 family proteins (note: the size of autophagosomes in HeLa cells is smaller than in MEFs, making it difficult to observe autophagosomes as ring-shaped structures). We have included this result in Figure 3–figure supplement 2(F) and explained it in the text (Lines 194–196, 198).

**Author response image 5. sa4fig5:** WT and ATG8 hexa KO HeLa cells stably expressing GFP–STX17TM and transiently expressing mRuby3–CERT(PHD) were cultured in starvation medium. Bars indicate 10 μm (main images) and 1 μm (insets).

(3) Can the authors try fusing Sac1 to one of the PI4P probes (CERT(PHD)) that were used, or alternatively to the c-terminus of Syntaxin 17? This approach would help to recruit Sac1 only to mature autophagosomes and could therefore prevent the autophagosome formation defect observed when fused to LC3B that targeted Sac1 to autophagosomes as they were forming. Understandably, this approach might seem a bit counterintuitive since the phosphatase is removing PI4P which is what is recruiting it but it could be a viable approach to keep PI4P levels low enough on mature autophagosomes so that Syntaxin 17 is no longer recruited. A Sac1 phosphatase mutant might be needed as a control.

We would like to thank the Reviewer for these suggestions. We tried the phosphatase domain of yeast Sac1 or human SAC1 fused with STX17TM, but unfortunately, these fusion proteins did not deplete PI4P from autophagosomes.